# PureGen: Universal Data Purification for Train-Time Poison Defense via Generative Model Dynamics

**Sunay Bhat** [†][∗]
sunaybhat1@ucla.edu

**Jeffrey Jiang** [†]
jeffrey.jiang@ucla.edu

**Omead Pooladzandi** [†]
opooladz@caltech.edu

**Alexander Branch**
alexrbranch@ucla.edu

**Gregory Pottie**
pottie@ee.ucla.edu

## Abstract

Train-time data poisoning attacks threaten machine learning models by introducing adversarial examples during training, leading to misclassification. Current defense methods often reduce generalization performance, are attack-specific, and impose significant training overhead. To address this, we introduce a set of universal data purification methods using a stochastic transform, $\Psi(x)$, realized via iterative Langevin dynamics of Energy-Based Models (EBMs), Denoising Diffusion Probabilistic Models (DDPMs), or both. These approaches purify poisoned data with minimal impact on classifier generalization. Our specially trained EBMs and DDPMs provide state-of-the-art defense against various attacks (including Narcissus, Bullseye Polytope, Gradient Matching) on CIFAR-10, Tiny-ImageNet, and CINIC-10, without needing attack or classifier-specific information. We discuss performance trade-offs and show that our methods remain highly effective even with poisoned or distributionally shifted generative model training data.

## 1 Introduction

Large datasets enable modern deep learning models but are vulnerable to data poisoning, where adversaries inject imperceptible poisoned images to manipulate model behavior at test time. Poisons can be created with or without knowledge of the model's architecture or training settings. As deep learning models grow in capability and usage, securing them against such attacks while preserving accuracy is critical.

Numerous methods of poisoning deep learning systems to create backdoors have been proposed in recent years. These disruptive techniques typically fall into two distinct categories: explicit backdoor, triggered data poisoning, or triggerless poisoning attacks. Triggered attacks conceal an imperceptible trigger pattern in the samples of the training data leading to the misclassification of test-time samples containing the hidden trigger [1, 2, 3, 4]. In contrast, triggerless poisoning attacks involve introducing slight, bounded perturbations to individual images that align them with target images of another class within the feature or gradient space resulting in the misclassification of specific instances without necessitating further modification during inference [5, 6, 7, 8, 9]. Alternatively, data availability attacks pose a training challenge by preventing model learning at train time, but do not introduce any latent backdoors that can be exploited at inference time [10, 11, 12]. In all these scenarios, poisoned examples often appear benign and correctly labeled making them challenging for observers or algorithms to detect.

---

[∗]Affiliation for all authors: University of California, Los Angeles
[†] Equal Contributions
GitHub: https://github.com/SunayBhat1/PureGen_PoisonDefense.

38th Conference on Neural Information Processing Systems (NeurIPS 2024).

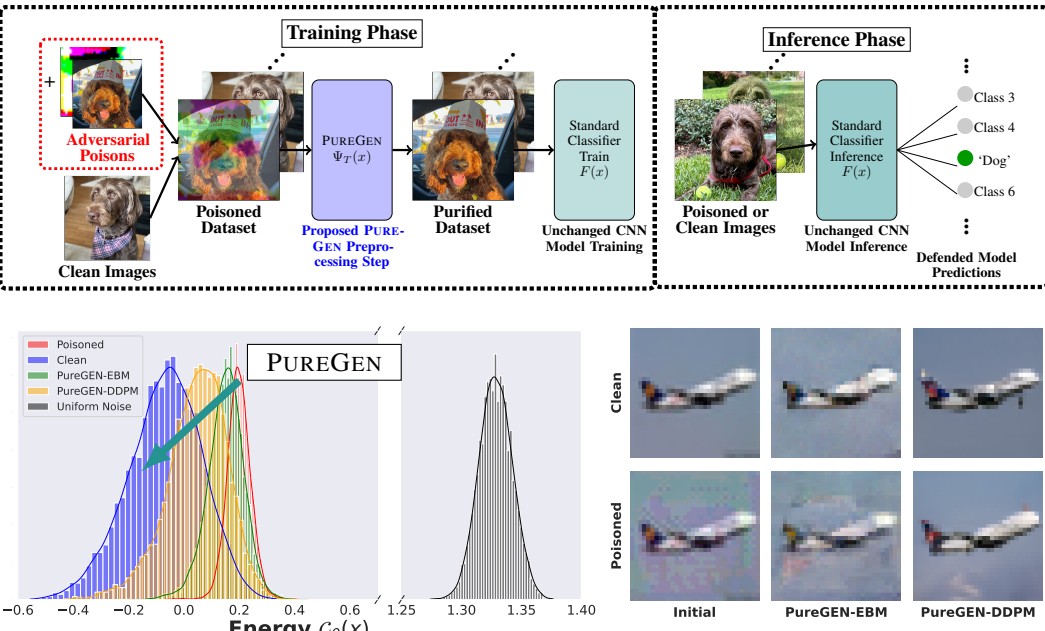

Figure 1: **Top** The full PUREGEN pipeline is shown where we apply our method as a preprocessing step with no further downstream changes to the classifier training or inference. *Poisoned images are moderately exaggerated to show visually.* **Bottom Left** Energy distributions of clean, poisoned, and PUREGEN purified images. Our methods push poisoned images via purification into the natural, clean image energy manifold. **Bottom Right** The removal of poison artifacts and the similarity of clean and poisoned images after purification using PUREGEN EBM and DDPM dynamics. The purified dataset results in SoTA defense and high classifier natural accuracy.

Current defense strategies against data poisoning exhibit significant limitations. While some methods rely on anomaly detection through techniques such as nearest neighbor analysis, training loss minimization, singular-value decomposition, feature activation or gradient clustering [13, 14, 15, 16, 17, 18, 19], others resort to robust training strategies including data augmentation, randomized smoothing, ensembling, adversarial training and maximal noise augmentation [20, 21, 22, 23, 24, 25, 26]. However, these approaches either undermine the model's generalization performance [27, 18], offer protection only against specific attack types [27, 17, 15], or prove computationally prohibitive for standard deep learning workflows [22, 16, 28, 18, 27, 17, 26]. There remains a critical need for more effective and practical defense mechanisms in the realm of deep learning security.

Generative models have been used for robust/adversarial training, but not for train-time backdoor attacks, to the best of our knowledge. Recent works have demonstrated the effectiveness of both EBM dynamics and Diffusion models to purify datasets against inference or availability attacks [29, 30, 31], but train-time backdoor attacks present additional challenges in both evaluation and application, requiring training using Public Out-of-Distribution (POOD) datasets and methods to avoid cumbersome computation or setup for classifier training.

We propose PUREGEN, a set of powerful stochastic preprocessing defense techniques, $\Psi_T(x)$, against train-time poisoning attacks. PUREGEN-EBM uses EBM-guided Markov Chain Monte Carlo (MCMC) sampling to purify poisoned images, while PUREGEN-DDPM uses a limited forward/reverse diffusion process, specifically for purification. Training DDPM models on a subset of the noise schedule improves purification by dedicating more model capacity to 'restoration' rather than generation. We further find that the energy of poisoned images is significantly higher than the baseline images, for a trained EBM, and PUREGEN techniques move poisoned samples to a lower-energy, natural data manifold with minimal accuracy loss. The PUREGEN pipeline, sample energy distributions, and purification on a sample image can be seen in Figure 1. PUREGEN significantly outperforms current defenses in all tested scenarios. Our key contributions in this work are as follows.

- A set of state-of-the-art (SoTA) stochastic preprocessing defenses $\Psi(x)$ against adversarial poisons using MCMC dynamics of EBMs and DDPMs trained specifically for purification

named PUREGEN-EBM and PUREGEN-DDPM with analysis providing further intuition on effectiveness

- Experimental results showing the broad application of $\Psi(x)$ with minimal tuning and no prior knowledge needed of the poison type and classification model
- Results showing SoTA performance can be maintained even when PUREGEN models' training data includes poisons or is from a significantly different distribution than the classifier/attacked train data distribution
- Results showing even further performance gains from combinations of PUREGEN-EBM and PUREGEN-DDPM and robustness to defense-aware poisons

## 2 Related Work

### 2.1 Targeted Data Poisoning Attack

Poisoning of a dataset occurs when an attacker injects small adversarial perturbations $\delta$ (where $\|\delta\|_\infty \leq \xi$ and typically $\xi = 8$ or $16/255$) into a small fraction, $\alpha$, of training images, making poisoning incredibly difficult to detect. These train-time attacks introduce *local sharp regions* with a considerably higher *training loss* [26]. A successful attack occurs when, after SGD optimizes the cross-entropy training objective on these poisoned datasets, invisible backdoor vulnerabilities are baked into a classifier, without a noticeable change in overall test accuracy. This is in contrast to inference-time or other adversarial scenarios where an attacker might be defense or model-aware. The goal in train-time attacks is "stealth" via minimal impact to the dataset and training and testing curves while creating backdoors to exploit at deployment.

In the realm of deep network poison security, we encounter two primary categories of attacks: triggered and triggerless attacks. Triggered attacks, often referred to as backdoor attacks, involve contaminating a limited number of training data samples with a specific trigger (often a patch) $\rho$ (similarly constrained $\|\rho\|_\infty \leq \xi$) that corresponds to a target label, $y^{\text{adv}}$. After training, a successful backdoor attack misclassifies when the perturbation $\rho$ is added:

$$F(x) = \begin{cases} y & x \in \{x : (x,y) \in \mathcal{D}_{test}\} \\ y^{\text{adv}} & x \in \{x + \rho : (x,y) \in \mathcal{D}_{test}, y \neq y^{\text{adv}}\} \end{cases} \tag{1}$$

Early backdoor attacks were characterized by their use of non-clean labels [32, 1, 33, 3], but more recent iterations of backdoor attacks have evolved to produce poisoned examples that lack a visible trigger [2, 34, 4].

On the other hand, triggerless poisoning attacks involve the addition of subtle adversarial perturbations to base images $\|\epsilon\|_\infty \leq \xi$, aiming to align their feature representations or gradients with those of target images of another class, causing target misclassification [5, 6, 7, 8, 9]. These poisoned images are virtually undetectable by external observers. Remarkably, they do not necessitate any alterations to the target images or labels during the inference stage. For a poison targeting a group of target images $\Pi = \{(x^\pi, y^\pi)\}$ to be misclassified as $y^{\text{adv}}$, an ideal triggerless attack would produce a resultant function:

$$F(x) = \begin{cases} y & x \in \{x : (x,y) \in \mathcal{D}_{test} \setminus \Pi\} \\ y^{\text{adv}} & x \in \{x : (x,y) \in \Pi\} \end{cases} \tag{2}$$

Background for data availability attacks can be found in [35]. We include results for one leading data availability attack Neural Tangent Gradient Attack (NTGA) [12], but we do not focus on such attacks since they are realized in model results during training. They do not pose a latent security risk in deployed models, and arguably have ethical applications within data privacy and content creator protections as discussed in App. 6.

The current leading poisoning attacks that we assess our defense against are listed below. More details about their generation can be found in App. A.1.

- **Bullseye Polytope (BP):** BP crafts poisoned samples that position the target near the center of their convex hull in a feature space [9].
- **Gradient Matching (GM):** GM generates poisoned data by approximating a bi-level objective by aligning the gradients of clean-label poisoned data with those of the adversarially-labeled target [8]. This attack has shown effectiveness against data augmentation and differential privacy.

- **Narcissus (NS):** NS is a clean-label backdoor attack that operates with minimal knowledge of the training set, instead using a larger natural dataset, evading state-of-the-art defenses by synthesizing persistent trigger features for a given target class. [4].
- **Neural Tangent Generalization Attacks (NTGA):** NTGA is a clean-label, black-box data availability attack that can collapse model test accuracy [12].

## 2.2 Train-Time Poison Defense Strategies

Poison defense categories broadly take two primary approaches: filtering and robust training techniques. Filtering methods identify outliers in the feature space through methods such as thresholding [14], nearest neighbor analysis [17], activation space inspection [16], or by examining the covariance matrix of features [15]. These defenses often assume that only a small subset of the data is poisoned, making them vulnerable to attacks involving a higher concentration of poisoned points. Furthermore, these methods substantially increase training time, as they require training with poisoned data, followed by computationally expensive filtering and model retraining [16, 17, 14, 15].

On the other hand, robust training methods involve techniques like randomized smoothing [20], extensive data augmentation [36], model ensembling [21], gradient magnitude and direction constraints [37], poison detection through gradient ascent [24], and adversarial training [27, 28, 25]. Additionally, differentially private (DP) training methods have been explored as a defense against data poisoning [22, 38]. Robust training techniques often require a trade-off between generalization and poison success rate [22, 37, 24, 28, 25, 26] and can be computationally intensive [27, 28]. Some methods use optimized noise constructed via Generative Adversarial Networks (GANs) or Stochastic Gradient Descent methods to make noise that defends against attacks [39, 26].

Recently Yang et al. [2022] proposed EPIC, a coreset selection method that rejects poisoned images that are isolated in the gradient space while training, and Liu et al. [2023] proposed FRIENDS, a per-image preprocessing transformation that solves a min-max problem to stochastically add $\ell_\infty$ norm $\zeta$-bound 'friendly noise' (typically 16/255) to combat adversarial perturbations (of 8/255) [18, 26].

These two methods are the previous SoTA and will serve as a benchmark for our PUREGEN methods in the experimental results. Finally, simple compression JPEG has been shown to defend against a variety of other adversarial attacks, and we apply it as a baseline defense in train-time poison attacks here as well, finding that it often outperforms previous SoTA methods [40].

## 3 PUREGEN: Purifying Generative Dynamics against Poisoning Attacks

### 3.1 Energy-Based Models and PUREGEN-EBM

An Energy-Based Model (EBM) is formulated as a Gibbs-Boltzmann density, as introduced in [41]. This model can be mathematically represented as:

$$p_\theta(x) = \frac{1}{Z(\theta)} \exp(-\mathcal{G}_\theta(x)) q(x), \tag{3}$$

where $x \in \mathcal{X} \subset \mathbb{R}^D$ denotes an image signal, and $q(x)$ is a reference measure, often a uniform or standard normal distribution. Here, $\mathcal{G}_\theta$ signifies the energy potential, parameterized by a ConvNet with parameters $\theta$.

The EBM $\mathcal{G}_\theta(x)$ can be interpreted as an unnormalized probability of how natural the image is to the dataset. Thus, we can use $\mathcal{G}_\theta(x)$ to filter images based on their likelihood of being poisoned. Furthermore, the EBM can be used as a generator. Given a starting clear or purified image $x_\tau$, we use Markov Chain Monte Carlo (MCMC) Langevin dynamics to iteratively generate more natural images via Equation 4.

$$x_{\tau+\Delta\tau} = x_\tau - \Delta\tau \nabla_{x_\tau} \mathcal{G}_\theta(x_\tau) + \sqrt{2\Delta\tau}\varepsilon_\tau, \tag{4}$$

where $\varepsilon_k \sim \mathcal{N}(0; \mathbf{I}_D)$, $\tau$ indexes the time step of the Langevin dynamics, and $\Delta\tau$ is the discretization of time [41]. $\nabla_x \mathcal{G}_\theta(x) = \partial \mathcal{G}_\theta(x)/\partial x$ can be obtained by back-propagation. Intuitively, the EBM informs a noisy stochastic gradient descent toward natural images. More details on the convergent contrastive learning mechanism of the EBM and mid-run generative dynamics that makes purification possible can be found in App. A.2.1. Ultimately, the training modifications of using realistic images

to initialize the MCMC runs of negative samples produces mid-run, meta-stable EBM dynamics which can be leveraged for better purification. Further intuition is in Section 3.4.

## 3.2 Diffusion Models and PUREGEN-DDPM

Denoising Diffusion Probabilistic Models (DDPMs) are a class of generative models proposed by [Ho et al., 2020] where the key idea is to define a forward diffusion process that adds noise until the image reaches a noise prior and then learn a reverse process that removes noise to generate samples as discussed further in App. A.3 [42]. For purification, we are interested in the stochastic "restoration" of the reverse process, where the forward process can degrade the image enough to remove adversarial perturbations. We find that only training the DDPM with a subset of the standard $\beta_t$ schedule, where the original image never reaches the prior, sacrifices generative capabilities for slightly improved poison defense while reducing training costs. Thus we introduce PUREGEN-DDPM which makes the simple adjustment of only training DDPMs for an initial portion of the standard forward process, improving purification capabilities. For our experiments, we find models trained up to 250 steps outperformed models in terms of poison purification than those trained on higher steps, up to the standard 1000 steps. We show visualizations and empirical evidence of this in Figure 2 below. In App. E.2.2 we show that pre-trained, standard DDPMs can offer comparable defense performance, but with added training cost.

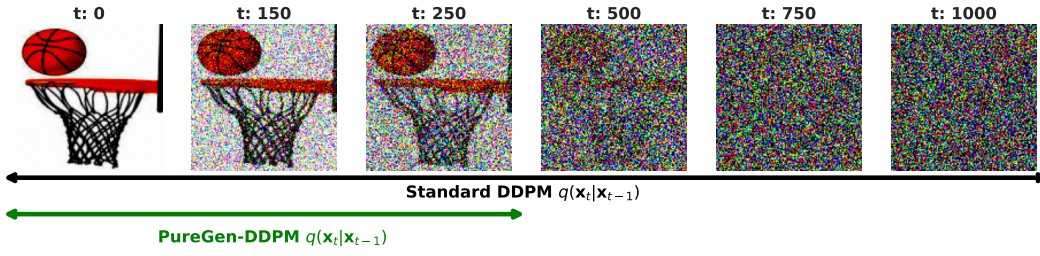

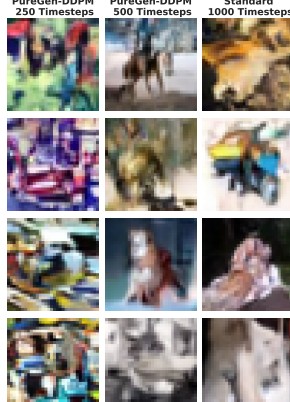

| Narcissus $\epsilon = 16$ 1% | | | | |
|---|---|---|---|---|
| *Purify Steps* | 75 | 100 | 125 | 150 |
| *Forward Train Steps* | Avg Natural Accuracy (%) ↑ | | | |
| 150 | $90.96 \pm 0.15$ | $90.21 \pm 0.20$ | $89.18 \pm 0.11$ | $88.46 \pm 0.22$ |
| **250** | $\mathbf{91.04 \pm 0.17}$ | $\mathbf{90.55 \pm 0.19}$ | $\mathbf{89.75 \pm 0.17}$ | $\mathbf{89.60 \pm 0.17}$ |
| 500 | $90.48 \pm 0.21$ | $89.77 \pm 0.20$ | $88.99 \pm 0.19$ | $88.19 \pm 0.15$ |
| 750 | $90.25 \pm 0.12$ | $89.06 \pm 0.18$ | $88.14 \pm 0.10$ | $87.19 \pm 0.21$ |
| 1000 | $90.11 \pm 0.16$ | $89.00 \pm 0.25$ | $87.98 \pm 0.18$ | $86.83 \pm 0.10$ |
| *Forward Train Steps* | Avg Poison Success (%) ↓ | | | |
| 150 | $8.03 \pm 6.36$ | $6.36 \pm 5.84$ | $5.51 \pm 4.07$ | $5.43 \pm 4.51$ |
| **250** | $\mathbf{7.14 \pm 6.94}$ | $\mathbf{5.58 \pm 5.25}$ | $\mathbf{4.36 \pm 3.63}$ | $\mathbf{4.15 \pm 3.24}$ |
| 500 | $8.88 \pm 7.31$ | $6.34 \pm 5.10$ | $5.45 \pm 4.22$ | $4.93 \pm 4.36$ |
| 750 | $9.27 \pm 6.26$ | $7.01 \pm 5.19$ | $5.96 \pm 4.64$ | $5.36 \pm 3.42$ |
| 1000 | $9.12 \pm 6.61$ | $7.01 \pm 4.82$ | $6.43 \pm 5.12$ | $5.12 \pm 3.18$ |

Figure 2: **Top** We compare PUREGEN-DDPM forward steps with the standard DDPM where 250 steps *degrades images for purification but does not reach a noise prior. Note that all model are trained with the same linear $\beta$ schedule.* **Bottom Left** Generated images from models with 250, 750, and 1000 (Standard) train forward steps *where it is clear 250 steps does not generate realistic images* **Bottom Right** Significantly improved poison defense performance of PUREGEN-DDPM with 250 train steps indicating a trade-off between data purification and generative capabilities.

## 3.3 Classification with Stochastic Transformation

Let $\Psi_T : \mathbb{R}^D \to \mathbb{R}^D$ be a stochastic pre-processing transformation. In this work, $\Psi_T(x)$, is the random variable of a fixed image $x$, and we define $T = (T_{\text{EBM}}, T_{\text{DDPM}}, T_{\text{Reps}}) \in \mathbb{R}^3$, hyperparameters specifying the number of EBM MCMC steps, the number of diffusion steps, and the number of times these steps are repeated, respectively. Then, $T_{\text{PUREGEN-EBM}} = (T_{\text{EBM}}, 0, 1)$ and $T_{\text{PUREGEN-DDPM}} = (0, T_{\text{DDPM}}, 1)$.

We compose a stochastic transformation $\Psi_{T,k}(x)$ with a randomly initialized deterministic classifier $f_\phi(x) \in \mathbb{R}^J$ (for us, a naturally trained classifier) to define a new deterministic classifier $F_\phi(x) \in \mathbb{R}^J$ as

$$F_\phi(x) = \mathbb{E}_{\Psi_{T,k}(x)}[f_{\phi_0}(\Psi_{T,k}(x))] \tag{5}$$

which is trained with cross-entropy loss via SGD to realize $F_\phi(x)$. As this is computationally infeasible we take $f_\phi(\Psi_{T,k}(x))$ as the point estimate of $F_\phi(x)$, which is valid because $\Psi_{T,k}(x)$ has low variance.

### 3.4 Erasing Poison Signals via Mid-Run MCMC

The stochastic transform $\Psi_T(x)$ is an iterative process. PUREGEN-EBM is akin to a noisy gradient descent over the unconditional energy landscape of a learned data distribution. This is more implicit in the PUREGEN-DDPM dynamics. As $T$ increases, poisoned images move from their initial higher energy towards more realistic lower-energy samples that lack poison perturbations. As shown in Figure 1, the energy distributions of poisoned images are much higher, pushing the poisons away from the likely manifold of natural images. By using Langevin dynamics of EBMs and DDPMs, we transport poisoned images back toward the center of the energy basin.

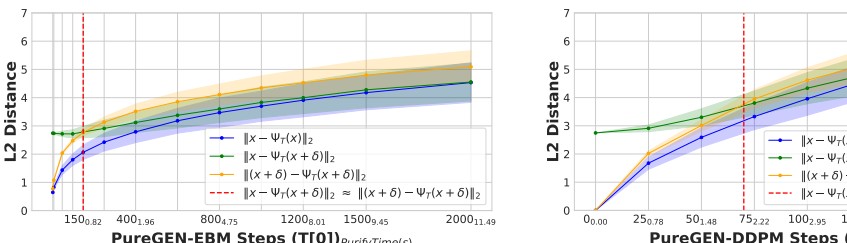

Figure 3: Plot of $\ell_2$ distances for PUREGEN-EBM (**Left**) and PUREGEN-DDPM (**Right**) between clean images and clean purified (blue), clean images and poisoned purified (green), and poisoned images and poisoned purified images (orange) at points on the Langevin dynamics trajectory. Purifying poisoned images for less than 250 steps moves a poisoned image closer to its clean image with a minimum around 150, preserving the natural image while removing the adversarial features.

In from-scratch $\epsilon = 8$ poison scenarios, 150 EBM Langevin steps or 75 DDPM steps fully purifies the majority of the dataset with minimal feature loss to the original image. In Figure 3, we explore the Langevin trajectory's impacts on $\ell_2$ distance of both purified clean and poisoned images from the initial clean image ($\|x - \Psi_T(x)\|_2$ and $\|x - \Psi_T(x+\delta)\|_2$), and the purified poisoned image's trajectory away from its poisoned starting point ($\|(x+\delta) - \Psi_T(x+\delta)\|_2$). Both poisoned and clean distance trajectories converge to similar distances away from the original clean image ($\lim_{T\to\infty} \|x - \Psi_T(x)\|_2 = \lim_{T\to\infty} \|x - \Psi_T(x+\delta)\|_2$), and the intersection where $\|(x+\delta) - \Psi_T(x+\delta)\|_2 > \|x - \Psi_T(x+\delta)\|_2$ (indicated by the dotted red line), occurs at ~150 EBM and 75 DDPM steps, indicating when purification has moved the poisoned image closer to the original clean image than the poisoned version of the image.

These dynamics provide a **concrete intuition for choosing step counts** that best balance poison defense with natural accuracy (given a poison $\epsilon$), hence why we use 150-1000 EBM steps of 75-125 (specifically 150 EBM, 75 DDPM steps in from-scratch scenarios) shown in App. D.2. Further, PUREGEN-EBM dynamics stay closer to the original images, while PUREGEN-DDPM moves further away as we increase the steps as the EBM has explicitly learned a probable data distribution, while the DDPM restoration is highly dependent on the conditional information in the degraded image. More experiments comparing the two are shown in App. G.2. These dynamics align with empirical results showing that EBMs better maintain natural accuracy and poison defense with smaller perturbations and across larger distributional shifts, but DDPM dynamics are better suited for larger poison perturbations. Finally, we note the purify times in the $x$-axes of Fig. 3, where PUREGEN-EBM is much faster for the same step counts to highlight the computational differences for the two methods, which we further explore Section 4.5.

Ultimately, one can think of PureGen as sampling from a "close" region in the pixel space around the original image where proximity is determined by a stochastic process that is initialized at the image

and remains "close" due to an explicit (EBM) or implicit (DDPM) energy gradient - all but assuring the original poison is mitigated in the process.

## 4 Experiments

### 4.1 Experimental Details

Table 1: Poison success and natural accuracy in all ResNet poison training scenarios. We report the mean and the standard deviations (as subscripts) of 100 GM experiments, 50 BP experiments, and NS triggers over 10 classes.

| | **From Scratch** | | | | | | | | | |
|---|---|---|---|---|---|---|---|---|---|---|
| | **CIFAR-10 (ResNet-18)** | | | | | **CINIC-10 (ResNet-18)** | | | **Tiny ImageNet (ResNet-34)** | |
| | **Gradient Matching-1%** | | **Narcissus-1%** | | | **Narcissus-1%** | | | **Gradient Matching-0.25%** | |
| | Poison Success (%)↓ | Avg Nat Acc (%)↑ | Avg Poison Success (%)↓ | Avg Nat Acc (%)↑ | Max Poison Success (%)↓ | Avg Poison Success (%)↓ | Avg Nat Acc (%)↑ | Max Poison Success (%)↓ | Poison Success (%)↓ | Avg Nat Acc (%)↑ |
| None | 44.00 | $94.84_{0.2}$ | $43.95_{33.6}$ | $94.89_{0.2}$ | 93.59 | $62.06_{0.21}$ | $86.32_{0.10}$ | 90.79 | 26.00 | $65.20_{0.5}$ |
| EPIC | 10.00 | $85.14_{1.2}$ | $27.31_{34.0}$ | $82.20_{1.1}$ | 84.71 | $49.50_{0.27}$ | $81.91_{0.08}$ | 91.35 | 18.00 | $60.55_{0.7}$ |
| FRIENDS | **0.00** | $91.15_{0.4}$ | $8.32_{22.3}$ | $91.01_{0.4}$ | 83.03 | $11.17_{0.25}$ | $77.53_{0.60}$ | 82.21 | 2.00 | $42.74_{7.5}$ |
| JPEG | **0.00** | $90.00_{0.19}$ | $1.78_{1.17}$ | $92.94_{0.15}$ | 4.13 | $18.89_{27.46}$ | $81.06_{0.18}$ | 92.12 | 10.00 | $60.01_{0.47}$ |
| **PUREGEN-DDPM** | **0.00** | $90.93_{0.20}$ | $1.64_{0.82}$ | $90.99_{0.22}$ | 2.83 | $4.76_{2.37}$ | $79.35_{0.08}$ | **7.74** | **0.00** | $50.50_{0.80}$ |
| **PUREGEN-EBM** | 1.00 | $92.98_{0.2}$ | $1.39_{0.8}$ | $92.92_{0.2}$ | **2.50** | $7.73_{0.08}$ | $82.37_{0.14}$ | 29.48 | 2.00 | $63.27_{0.4}$ |
| | **Transfer Learning (CIFAR-10, ResNet-18)** | | | | | | | | | |
| | **Fine-Tune** | | | | | **Linear - Bullseye Polytope** | | | | |
| | **Bullseye Polytope-10%** | | **Narcissus-10%** | | | **BlackBox-10%** | | **WhiteBox-1% (CIFAR-100)** | | |
| | Poison Success (%)↓ | Avg Nat Acc (%)↑ | Avg Poison Success (%)↓ | Avg Nat Acc (%)↑ | Max Poison Success (%)↓ | Poison Success (%)↓ | Avg Nat Acc (%)↑ | Poison Success (%)↓ | Avg Nat Acc (%)↑ | |
| None | 46.00 | $89.84_{0.9}$ | $33.41_{33.9}$ | $90.14_{2.4}$ | 98.27 | 93.75 | $83.59_{2.4}$ | 98.00 | $70.09_{0.2}$ | |
| EPIC | 42.00 | $81.95_{5.6}$ | $20.93_{27.1}$ | $88.58_{2.0}$ | 91.72 | 66.67 | $84.34_{3.8}$ | 63.00 | $60.86_{1.5}$ | |
| FRIENDS | 8.00 | $87.82_{1.2}$ | $3.04_{5.1}$ | $89.81_{0.5}$ | 17.32 | 33.33 | $85.18_{2.3}$ | 19.00 | $60.90_{0.6}$ | |
| JPEG | **0.00** | $90.40_{0.44}$ | $2.95_{3.71}$ | $87.63_{0.49}$ | 12.55 | **0.00** | $92.44_{0.47}$ | 8.00 | $50.42_{0.73}$ | |
| **PUREGEN-DDPM** | **0.00** | $91.53_{0.15}$ | $1.88_{1.12}$ | $90.69_{0.26}$ | **3.42** | **0.00** | $93.81_{0.08}$ | 9.0 | $54.53_{0.64}$ | |
| **PUREGEN-EBM** | **0.00** | $87.52_{1.2}$ | $2.02_{1.0}$ | $89.78_{0.6}$ | 3.85 | **0.00** | $92.38_{0.3}$ | **6.00** | $64.98_{0.3}$ | |

We evaluate PUREGEN-EBM and PUREGEN-DDPM against state-of-the-art defenses EPIC and FRIENDS, and baseline defense JPEG, on leading poisons Narcissus (NS), Gradient Matching (GM), and Bullseye Polytope (BP). Using ResNet18 and CIFAR-10, we measure poison success, natural accuracy, and max poison success across classes for triggered NS attacks. All poisons and poison scenario settings come from previous baseline attack and defense works, and additional details on poison sources, poison crafting, definitions of poison success, and training hyperparameters can be found in App. D. Poisons were chosen for their availability or ease of generation from the poison-crafting research community, which is why there are no GM results on CINIC-10 and no Narcissus results on Tiny-ImageNet. And we note that certain poison successes (GM and BP) are for moving a single image to a target class per 50-100 classifier scenarios and, hence, lack a standard deviation. Athough, we show the results are low variance using different seeds on a subset of scenarios in App. E.2.3.

Our EBMs and DDPMs are trained on the ImageNet (70k) portion of the CINIC-10 dataset, CIFAR-10, and CalTech-256 for poisons scenarios using CIFAR-10, CINIC-10, and Tiny-ImageNet respectively, to ensure no overlap of PUREGEN train and attacked classifier train datasets [43, 44, 45].

### 4.2 Benchmark Results

Table 1 shows our primary results using ResNet18 (*34 for Tiny-IN*) in which PUREGEN achieves state-of-the-art (SoTA) poison defense and natural accuracy in all poison scenarios. Both PUREGEN methods show large improvements over baselines in triggered NS attacks (PUREGEN matches or exceeds previous SoTA with a 1-6% poison defense reduction and 0.5-1.5% less degradation in natural accuracy), while maintaining perfect or near-perfect defense with improved natural accuracy in triggerless BP and GM scenarios. Note that PUREGEN-EBM does a better job maintaining natural accuracy in the 100 class scenarios (BP-WhiteBox and Tiny-IN), while PUREGEN-DDPM tends to get much better poison defense when the PUREGEN-EBM is not already low.

Table 2 shows selected results for additional models MobileNetV2, DenseNet121, and Hyperlight Benchmark (HLB), which is a unique case study with a residual-less network architecture, unique initialization scheme, and super-convergence training method that recently held the world record of achieving 94% test accuracy on CIFAR-10 with just 10 epochs [46]. Due to the fact that PUREGEN is a preprocessing step, it can be readily applied to novel training paradigms with no modifications

Table 2: Results for additional models (MobileNetV2, DenseNet121, and HLB) and the NTGA data-availability attack. PUREGEN remains state-of-the-art for all train-time *latent* attacks, while NTGA defense shows near SoTA performance. *All NTGA baselines pulled from [30].

| Scenario | From Scratch NS($\epsilon = 8$)-1% | | | | | | Linear Transfer BlackBox BP-10% | | | |
|---|---|---|---|---|---|---|---|---|---|---|
| Model | MobileNetV2 | | DenseNet121 | | Hyperlight Bench | | MobileNetV2 | | DenseNet121 | |
| Defense | Avg Poison Success (%) ↓ | Avg Nat Acc (%) ↑ | Avg Poison Success (%) ↓ | Avg Nat Acc (%) ↑ | Avg Poison Success (%) ↓ | Avg Nat Acc (%) ↑ | Poison Success (%) ↓ | Avg Nat Acc (%) ↑ | Poison Success (%) ↓ | Avg Nat Acc (%) ↑ |
| None | $32.70_{0.25}$ | $93.92_{0.13}$ | $46.52_{32.2}$ | $95.33_{0.1}$ | $76.39_{16.35}$ | $93.95_{0.10}$ | $81.25$ | $73.27_{0.97}$ | $73.47$ | $82.13_{1.62}$ |
| EPIC | $22.35_{0.24}$ | $78.16_{9.93}$ | $32.60_{29.4}$ | $85.12_{2.4}$ | $10.58_{18.35}$ | $24.88_{6.04}$ | $56.25$ | $54.47_{5.57}$ | $66.67$ | $70.20_{10.15}$ |
| FRIENDS | $\mathbf{2.00}_{0.01}$ | $88.82_{0.57}$ | $8.60_{21.2}$ | $91.55_{0.3}$ | $11.35_{18.45}$ | $87.03_{1.52}$ | $41.67$ | $68.86_{1.50}$ | $60.42$ | $80.22_{1.90}$ |
| JPEG | $2.30_{1.20}$ | $86.60_{0.13}$ | $\mathbf{1.90}_{1.54}$ | $92.03_{0.22}$ | $\mathbf{1.73}_{0.97}$ | $90.92_{0.22}$ | $2.08$ | $73.14_{0.71}$ | $0.00$ | $78.67_{1.60}$ |
| PUREGEN-DDPM | $2.13_{1.02}$ | $86.91_{0.23}$ | $1.71_{0.94}$ | $90.94_{0.23}$ | $1.75_{0.90}$ | $89.34_{0.25}$ | $0.00$ | $\mathbf{83.15}_{0.02}$ | $0.00$ | $\mathbf{89.02}_{0.15}$ |
| PUREGEN-EBM | $\mathbf{1.64}_{0.01}$ | $\mathbf{91.75}_{0.13}$ | $\mathbf{1.42}_{0.7}$ | $\mathbf{93.48}_{0.1}$ | $1.89_{1.06}$ | $\mathbf{91.94}_{0.14}$ | $0.00$ | $78.57_{1.37}$ | $0.00$ | $89.29_{0.94}$ |

| NTGA Data Availability Attack | | | | | | | | | |
|---|---|---|---|---|---|---|---|---|---|
| Defense | None | FAutoAug.* | Median Blur* | TVM* | Grayscale* | AVATAR * | JPEG | PUREGEN-DDPM | PUREGEN-EBM |
| Avg Natural Accuracy (%) ↑ | $11.49_{0.69}$ | $27.56_{2.45}$ | $28.43_{1.41}$ | $41.41_{1.37}$ | $\mathbf{81.27}_{0.27}$ | $\mathbf{86.22}_{0.38}$ | $79.22_{0.25}$ | $83.48_{0.43}$ | $85.22_{0.38}$ |

unlike previous baselines EPIC and FRIENDS. In all results, PUREGEN is again SoTA, except for NTGA data-availability attack, where PUREGEN is just below SoTA method AVATAR (which is also a diffusion based approach). But we again emphasize data-availability attacks are not the focus of PUREGEN which secures against latent attacks.

The complete results for all models and all versions of each baseline can be found in App. E.

## 4.3 PUREGEN Robustness to Train Data Shifts, Poisoning, and Defense-Aware Poisons

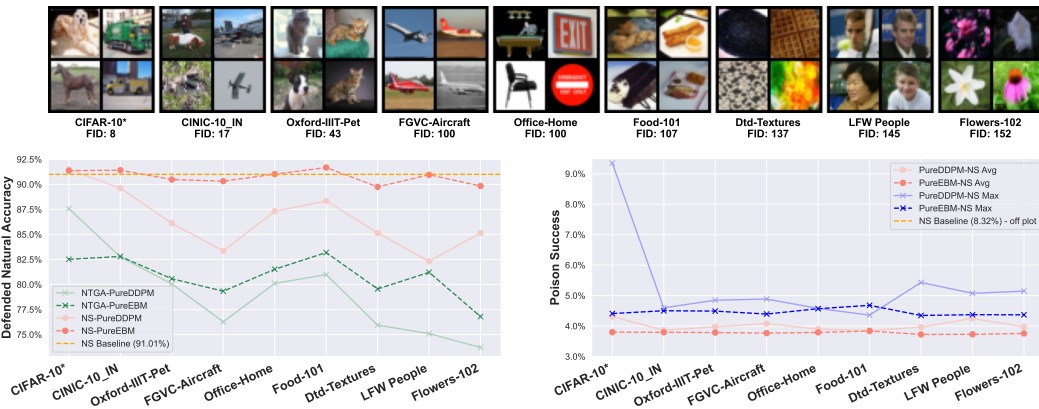

Figure 4: PUREGEN-EBM vs. PUREGEN-DDPM with increasingly Out-of-Distribution training data (for generative model training) and purifying target/attacked distribution CIFAR-10. PUREGEN-EBM is much more robust to distributional shift for natural accuracy while both PUREGEN-EBM and PUREGEN-DDPM maintain SoTA poison defense across all train distributions *CIFAR-10 is a "cheating" baseline as clean versions of poisoned images are present in training data.*

An important consideration for PUREGEN is the distributional shift between the data used to train the generative models and the target dataset to be purified. Figure 4 explores this by training PUREGEN-EBM and PUREGEN-DDPM on increasingly out-of-distribution (OOD) datasets while purifying the CIFAR-10 dataset (NS attack). We quantify the distributional shift using the Fréchet Inception Distance (FID) [47] between the original CIFAR-10 training images and the OOD datasets. Notably, both methods maintain SoTA or near SoTA poison defense across all training distributions, highlighting their effectiveness even under distributional shift. The results show that PUREGEN-EBM is more robust to distributional shift in terms of maintaining natural accuracy, with only a slight drop in performance even when trained on highly OOD datasets like Flowers-102 and LFW people. In contrast, PUREGEN-DDPM experiences a more significant drop in natural accuracy as the distributional shift increases. Note that the CIFAR-10 is a "cheating" baseline, as clean versions of the poisoned images are present in the generative model training data, but it provides an upper bound on the performance that can be achieved when the generative models are trained on perfectly in-distribution data.

Table 3: Both PUREGEN-EBM and PUREGEN-DDPM are robust to NS attack even when fully poisoned (*all classes at once*) during model training except for NS Eps-16 for PUREGEN-EBM

| *Classifier NS Attack Eps* | | 8 | | | 16 | | |
| *PureGen w/NS Training Poison* | Nat Acc (%) ↑ | Poison Success (%) ↓ | Max Poison (%) ↓ | Nat Acc (%) ↑ | Poison Success (%) ↓ | Max Poison (%) ↓ |
|---|---|---|---|---|---|---|
| 0 **PUREGEN-DDPM** | $91.51_{0.13}$ | $2.62_{3.75}$ | **12.70** | $90.31_{0.18}$ | $4.61_{3.99}$ | **12.86** |
|    PUREGEN-EBM | $91.37_{0.14}$ | $1.60_{0.82}$ | **2.82** | $88.21_{0.15}$ | $8.73_{6.29}$ | 23.05 |
| 8 **PUREGEN-DDPM** | $88.99_{0.16}$ | $1.65_{0.79}$ | **2.87** | $85.24_{0.10}$ | $4.79_{2.83}$ | 10.53 |
|    PUREGEN-EBM | $91.11_{0.18}$ | $1.55_{0.89}$ | **2.87** | $87.60_{0.18}$ | $5.35_{3.30}$ | 12.05 |
| 16 **PUREGEN-DDPM** | $88.02_{0.21}$ | $1.57_{0.79}$ | **2.79** | $83.74_{0.21}$ | $2.90_{1.54}$ | **6.11** |
|    PUREGEN-EBM | $90.76_{0.14}$ | $1.28_{0.86}$ | **3.43** | $85.58_{0.40}$ | $17.73_{14.62}$ | 44.15 |

Another important consideration is the robustness of PUREGEN when the generative models themselves are trained on poisoned data. Table 3 shows the performance of PUREGEN-EBM and PUREGEN-DDPM when their training data is fully poisoned with the Narcissus (NS) attack, *meaning that all classes are poisoned simultaneously.* The results demonstrate that both PUREGEN-EBM and PUREGEN-DDPM are highly robust to poisoning during model training, maintaining SoTA poison defense and natural accuracy with only exception being PUREGEN-EBM's performance on the more challenging NS $\epsilon = 16$ attack when poisoned with the same perturbations. While it is unlikely an attacker would have access to both the the generative model and classifier train datasets, these findings highlight the inherent robustness of PUREGEN, as the generative models can effectively learn the underlying clean data distribution even in the presence of poisoned samples during training. This is a key advantage of PUREGEN compared to other defenses, especially when there is no secure dataset.

Finally, in App. E.2.1, we show results where we integrate an EBM into the Narcissus crafting pipeline, done by taking a gradient through the EBM MCMC dynamics in three ways based on the specifics of crafting. In all cases, PUREGEN shows almost no defense degradation, and we actually show we can generate more effective poisons over baseline this way. These results further validate the effectiveness of PUREGEN even with defense-aware crafting, which is not a typical assumption in train-time attacks.

## 4.4 PUREGEN Extensions on Higher Power Attacks

To leverage the strengths of both PUREGEN-EBM and PUREGEN-DDPM, we propose PUREGEN combinations that utilize either both EMB and DPPM back-to-back (PUREGEN-NAIVE), EBM and DDPM with multiple repetitions of smaller steps (PUREGEN-REPS), and finally EBM as a filter to then use EBM/DDPM on only the k highest energy samples as described in 3.3. For additional description, see C, and note that these extensions required extensive hyperparameter search with performance sweeps shown in App F, as there was little intuition for the amount of reps ($T_{Reps}$) or the filtering threshold ($k$) needed. Thus, we do not include these methods in our core results, but we do show the added performance gains on higher power poisons in Table 4, both in terms of increased perturbation size $\epsilon = 16$ and increased poison % (and both together).

Table 4: PUREGEN-NAIVE, PUREGEN-REPS, and PUREGEN-FILT results showing further performance gains on increased poison power scenarios

| | Narcissus $\epsilon = 8$ **10%** | | | Narcissus $\epsilon = 16$ **1%** | | | Narcissus $\epsilon = 16$ **10%** | | |
| | Avg Poison Success (%) ↓ | Avg Nat Acc (%) ↑ | Max Poison Success (%) ↓ | Avg Poison Success (%) ↓ | Avg Nat Acc (%) ↑ | Max Poison Success (%) ↓ | Avg Poison Success (%) ↓ | Avg Nat Acc (%) ↑ | Max Poison Success (%) ↓ |
|---|---|---|---|---|---|---|---|---|---|
| None | $96.27_{6.62}$ | $84.57_{0.60}$ | 99.97 | $83.63_{12.09}$ | $93.67_{0.11}$ | 97.36 | $99.35_{0.81}$ | $84.58_{0.63}$ | 99.97 |
| Best Baseline | $16.95_{9.72}$ | $84.66_{1.51}$ | 33.96 | $11.85_{12.60}$ | $87.72_{0.19}$ | 36.90 | $71.28_{22.90}$ | $82.83_{0.43}$ | 99.25 |
| **PUREGEN-DDPM** | $6.38_{5.16}$ | $85.86_{0.46}$ | 16.29 | $5.21_{3.35}$ | $86.16_{0.19}$ | 13.32 | $69.38_{16.73}$ | $83.58_{1.02}$ | 89.35 |
| **PUREGEN-EBM** | $52.48_{23.29}$ | $86.14_{1.82}$ | 99.86 | $7.35_{4.46}$ | $85.61_{0.25}$ | 16.94 | $77.50_{7.01}$ | $78.79_{0.93}$ | 90.84 |
| **PUREGEN-NAIVE** | $10.43_{8.58}$ | $88.20_{0.54}$ | 27.42 | $5.20_{2.61}$ | $85.95_{0.23}$ | **9.80** | $63.01_{15.24}$ | $83.14_{0.90}$ | 87.17 |
| **PUREGEN-REPS** | $3.75_{2.28}$ | $85.56_{0.22}$ | **7.74** | $4.95_{2.48}$ | $85.79_{0.18}$ | 10.75 | $53.79_{17.14}$ | $83.92_{1.02}$ | **81.09** |
| **PUREGEN-FILT** | $6.47_{6.98}$ | $86.08_{2.00}$ | 18.81 | $5.74_{4.05}$ | $90.52_{0.18}$ | 16.08 | $69.13_{12.94}$ | $85.47_{1.45}$ | 87.66 |

## 4.5 PUREGEN Timing and Limitations

Table 5 presents the training times for using PUREGEN-EBM and PUREGEN-DDPM (923 and 4181 seconds respectively to purify) on CIFAR-10 using a TPU V3. Although these times may seem significant, PUREGEN is a universal defense applied once per dataset, making its cost negligible when reused across multiple tasks and poison scenarios. To highlight this, we also present the purification times amortized over the 10 and 100 NS and GM poison scenarios, demonstrating that

the cost becomes negligible when the purified dataset is used multiple times relative to baselines like FRIENDS which require retraining for each specific task and poison scenario (while still utilizing the full dataset unlike EPIC). PUREGEN-EBM generally has lower purification times compared to PUREGEN-DDPM, making it more suitable for subtle and rare perturbations. Conversely, PUREGEN-DDPM can handle more severe perturbations but at a higher computational cost and potential reduction in natural accuracy.

Table 5: PUREGEN and baseline Timing Analysis on TPU V3

| | Train Time (seconds) | | |
| --- | --- | --- | --- |
| | Single Classifier (Median) | Gradient Matching 100 Classifiers | Narcissus 10 Classifiers |
| None, JPEG | **3690** | $3673_{48}$ | **$5288_{29}$** |
| EPIC | 3650 | $3624_{153}$ | $5212_{295}$ |
| FRIENDS | 11502 | $11578_{627}$ | $128668_{5573}$ |
| PUREGEN-EBM $_{T=[150,0,1]}$ | 4613 | $3699_{48}$ | **$5380_{32}$** |
| PUREGEN-DDPM $_{T=[0,75,1]}$ | 7871 | $3731_{48}$ | $5706_{37}$ |

Training the generative models for PUREGEN involves substantial computational cost and data requirements. However, as shown in Table 3 and Figure 4, these models remain effective even when trained on poisoned or out-of-distribution data. This universal applicability justifies the initial training cost, as the models can defend against diverse poisoning scenarios. So while JPEG is a fairly effective baseline, the added benefits of PUREGEN start to outweigh the compute as the use cases of the dataset increase. While PUREGEN combinations (PUREGEN-REPS and PUREGEN-FILT) show enhanced performance on higher power attacks (Table 4), further research is needed to fully exploit the strengths of both PUREGEN-EBM and PUREGEN-DDPM.

## 5  Conclusion

Poisoning has the potential to become one of the greatest attack vectors to AI models, decreasing model security and eroding public trust. In this work, we introduce PUREGEN, a suite of universal data purification methods that leverage the stochastic dynamics of Energy-Based Models (EBMs) and Denoising Diffusion Probabilistic Models (DDPMs) to defend against train-time data poisoning attacks. PUREGEN-EBM and PUREGEN-DDPM effectively purify poisoned datasets by iteratively transforming poisoned samples into the natural data manifold, thus mitigating adversarial perturbations. Our extensive experiments demonstrate that these methods achieve state-of-the-art performance against a range of leading poisoning attacks and can maintain SoTA performance in the face of poisoned or distributionally shifted generative model training data. These versatile and efficient methods set a new standard in protecting machine learning models against evolving data poisoning threats, potentially inspiring greater trust in AI applications.

## 6  Potential Social Impacts

Poisoning represents one of the greatest emerging threats to AI systems, particularly as foundation models increasingly rely on large, diverse datasets without rigorous quality control against imperceptible perturbations. This vulnerability is especially concerning in high-stakes domains like healthcare, security, and autonomous vehicles, where model integrity is crucial and erroneous outputs could have catastrophic consequences. Our research provides a universal defense method that can be implemented with minimal impact to existing training infrastructure, enabling practitioners to preemptively secure their datasets against state-of-the-art poisoning attacks.

While we acknowledge that the poison defense space can promote an 'arms race' of increasingly sophisticated attacks and defenses, our approach's universality poses a fundamentally harder challenge for attackers, even when using defense-aware crafting E.2.1. We specifically focus on defending against latent backdoor vulnerabilities rather than data availability attacks, as the latter can serve legitimate purposes in protecting content creators' rights. By providing robust defense against malicious poisoning while preserving natural model performance, our method helps build trust in AI systems for increasingly consequential real-world applications.

## 7 Acknowledgments

This work is supported with Cloud TPUs from Google's Tensorflow Research Cloud (TFRC). We would like to acknowledge Jonathan Mitchell, Mitch Hill, Yuan Du and Kathrine Abreu for support and discussion on base EBM and Diffusion code, and Yunzheng Zhu for his help in crafting poisons.

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

# A  Further Background

## A.1  Poisons

The goal of adding train-time poisons is to change the prediction of a set of target examples $\Pi = \{(x^\pi, y^\pi)\} \subset \mathcal{D}_{test}$ or triggered examples $\{(x + \rho, y) : (x, y) \in \mathcal{D}_{test}\}$ to an adversarial label $y^{\text{adv}}$.

Targeted clean-label data poisoning attacks can be formulated as the following bi-level optimization problem:

$$
\underset{\substack{\delta_i \in \mathcal{C}_\delta, \rho \in \mathcal{C}_\rho \\ \sum_{i=0}^n \mathbb{1}_{\delta_i \neq \mathbf{0}} \leq \alpha n}}{\arg\min} \sum_{(x^\pi, y^\pi) \in \Pi} \mathcal{L}\left(F(x^\pi + \rho; \phi(\delta)), y^{\text{adv}}\right)
$$

$$
s.t. \quad \phi(\delta) = \underset{\phi}{\arg\min} \sum_{(x,y) \in \mathcal{D}} \mathcal{L}\left(F(x + \delta_i; \phi), y\right) \tag{6}
$$

For a triggerless poison, we solve for the ideal perturbations $\delta_i$ to minimize the adversarial loss on the target images, where $\mathcal{C}_\delta = \mathcal{C}$, $\mathcal{C}_\rho = \{\mathbf{0} \in \mathbb{R}^D\}$, and $\mathcal{D} = \mathcal{D}_{train}$. To address the above optimization problem, powerful poisoning attacks such as Meta Poison (MP) [7], Gradient Matching (GM) [8], and Bullseye Polytope (BP) [9] craft the poisons to mimic the gradient of the adversarially labeled target, i.e.,

$$
\nabla \mathcal{L}\left(F_\phi\left(x^\pi\right), y^{\text{adv}}\right) \propto \sum_{i : \delta_i \neq \mathbf{0}} \nabla \mathcal{L}\left(F_\phi(x_i + \delta_i), y_i\right) \tag{7}
$$

Minimizing the training loss on RHS of Equation 7 also minimizes the adversarial loss objective of Equation 6.

For the triggered poison, Narcissus (NS), we find the most representative patch $\rho$ for class $\pi$ given $\mathcal{C}$, defining Equation 6 with $\mathcal{C}_\delta = \{\mathbf{0} \in \mathbb{R}^D\}$, $\mathcal{C}_\rho = \mathcal{C}$, $\Pi = \mathcal{D}_{train}^\pi$, $y^{\text{adv}} = y^\pi$, and $\mathcal{D} = \mathcal{D}_{POOD} \cup \mathcal{D}_{train}^\pi$. In particular, this patch uses a public out-of-distribution dataset $\mathcal{D}_{POOD}$ and only the targeted class $\mathcal{D}_{train}^\pi$. As finding this patch comes from another natural dataset and does not depend on other train classes, NS has been more flexible to model architecture, dataset, and training regime [4].

Background for data availability attacks can be found in [35]. The goal for data availability attacks (sometimes referred to as "unlearnable" attacks is to collapse the test accuracy, and hence the model's ability to generalize, or learn useful representations, from the train dataset. As we discuss in the main paper, such attacks are immediately obvious when training a model, or rather create a region of poor performance in the model. These attacks do not create latent vulnerabilities that can then be exploited by an adversary. Thus we do not focus on, or investigate our methods with any detail for availability attacks. Further, we discuss in the Societal Impacts section how such attacks have many ethical uses for privacy and data protection 6.

## A.2  Further EBM Discussions

Recalling the Gibbs-Boltzmann density from Equation 3,

$$
p_\theta(x) = \frac{1}{Z(\theta)} \exp(-\mathcal{G}_\theta(x)) q(x), \tag{8}
$$

where $x \in \mathcal{X} \subset \mathbb{R}^D$ denotes an image signal, and $q(x)$ is a reference measure, often a uniform or standard normal distribution. Here, $\mathcal{G}_\theta$ signifies the energy potential, parameterized by a ConvNet with parameters $\theta$.

The normalizing constant, or the partition function, $Z(\theta) = \int \exp\{-\mathcal{G}_\theta(x)\} q(x) dx = \mathbb{E}_q[\exp(-\mathcal{G}_\theta(x))]$, while essential, is generally analytically intractable. In practice, $Z(\theta)$ is not computed explicitly, as $\mathcal{G}_\theta(x)$ sufficiently informs the Markov Chain Monte Carlo (MCMC) sampling process.

As which $\alpha$ of the images are poisoned is unknown, we treat them all the same for a universal defense. Considering i.i.d. samples $x_i \sim \mathbb{P}$ for $i = 1, \ldots, n$, with $n$ sufficiently large, the sample average over $x_i$ converges to the expectation under $\mathbb{P}$ and one can learn a parameter $\theta^*$ such that $p_{\theta^*}(x) \approx \mathbb{P}(x)$. For notational simplicity, we equate the sample average with the expectation.

The objective is to minimize the expected negative log-likelihood, formulated as:

$$\mathcal{L}(\theta) = \frac{1}{n} \sum_{i=1}^{n} \log p_\theta(x_i) \doteq \mathbb{E}_{\mathbb{P}}[\log p_\theta(x)]. \tag{9}$$

The derivative of this log-likelihood, crucial for parameter updates, is given by:

$$\nabla_\theta \mathcal{L}(\theta) = \mathbb{E}_{\mathbb{P}}\left[\nabla_\theta \mathcal{G}_\theta(x)\right] - \mathbb{E}_{p_\theta}\left[\nabla_\theta \mathcal{G}_\theta(x)\right]$$

$$\doteq \frac{1}{n} \sum_{i=1}^{n} \nabla_\theta \mathcal{G}_\theta(x_i^+) - \frac{1}{k} \sum_{i=1}^{k} \nabla_\theta \mathcal{G}_\theta(x_i^-), \tag{10}$$

where solving for the critical points results in the average gradient of a batch of real images $(x_i^+)$ should be equal to the average gradient of a synthesized batch of examples from the real images $x_i^- \sim p_\theta(x)$. The parameters are then updated as $\theta_{t+1} = \theta_t + \eta_t \nabla \mathcal{L}(\theta_t)$, where $\eta_t$ is the learning rate.

In this work, to obtain the synthesized samples $x_i^-$ from the current distribution $p_\theta(x)$ we use the iterative application of the Langevin update as the Monte Carlo Markov Chain (MCMC) method, first introduced in Equation 4:

$$x_{\tau+\Delta\tau} = x_\tau - \Delta\tau \nabla_{x_\tau} \mathcal{G}_\theta(x_\tau) + \sqrt{2\Delta\tau}\epsilon_\tau, \tag{11}$$

where $\epsilon_\tau \sim N(0, I_D)$, $\tau$ indexes the time step of the Langevin dynamics, and $\Delta\tau$ is the discretization of time [41]. $\nabla_x \mathcal{G}_\theta(x) = \partial \mathcal{G}_\theta(x)/\partial x$ can be obtained by back-propagation. If the gradient term dominates the diffusion noise term, the Langevin dynamics behave like gradient descent. We implement EBM training following [48], see App. A.2.1 for details.

---

**Algorithm 1** Data Preprocessing with PUREGEN-EBM: $\Psi_T(x)$

---

**Require:** Trained ConvNet potential $\mathcal{G}_\theta(x)$, training images $x \in X$, Langevin steps $T$, Time discretization $\Delta\tau$
    **for** $\tau$ in $1 \ldots T$ **do**
        Langevin Step: draw $\epsilon_\tau \sim N(0, I_D)$

$$x_{\tau+1} = x_\tau - \Delta\tau \nabla_{x_\tau} \mathcal{G}_\theta(x_\tau) + \sqrt{2\Delta\tau}\epsilon_\tau$$

    **end for**
**Return:** Purified set $\tilde{X}$ from final Langevin updates

---

### A.2.1 EBM Training

Algorithm 2 is pseudo-code for the training procedure of a data-initialized convergent EBM. We use the generator architecture of the SNGAN [49] for our EBM as our network architecture. Further intuiton can be found in App. B.1.

### A.3 DDPM Background

The forward process successively adds noise over a sequence of time steps, eventually resulting in values that follow a prior distribution, typically a standard Gaussian as in:

$$q(\boldsymbol{x}_t|\boldsymbol{x}_{t-1}) = \mathcal{N}(\boldsymbol{x}_t; \sqrt{1-\beta_t}\boldsymbol{x}_{t-1}, \beta_t\mathbf{I}) \tag{12}$$

where $\boldsymbol{x}_0 \sim q(\boldsymbol{x}_0)$ be a clean image sampled from the data distribution. The forward process is defined by a fixed Markov chain with Gaussian transitions for a sequence of timesteps $t = 1, \ldots, T$:

The reverse process is defined as the conditional distribution of the previous variable at a timestep, given the current one:

$$p_\theta(\boldsymbol{x}_{t-1}|\boldsymbol{x}_t) = \mathcal{N}(\boldsymbol{x}_{t-1}; \mu_\theta(\boldsymbol{x}_t, t), \Sigma_\theta(\boldsymbol{x}_t, t)) \tag{13}$$

---

**Algorithm 2** ML with SGD for Convergent Learning of EBM (3)

---

**Require:** ConvNet potential $\mathcal{G}_\theta(x)$, number of training steps $J = 150000$, initial weight $\theta_1$, training images $\{x_i^+\}_{i=1}^{N_{\text{data}}}$, data perturbation $\tau_{\text{data}} = 0.02$, step size $\tau = 0.01$, Langevin steps $T = 100$, SGD learning rate $\gamma_{\text{SGD}} = 0.00005$.

**Ensure:** Weights $\theta_{J+1}$ for energy $\mathcal{G}_\theta(x)$.

Set optimizer $g \leftarrow \text{SGD}(\gamma_{\text{SGD}})$. Initialize persistent image bank as $N_{\text{data}}$ uniform noise images.
**for** $j$=1:($J$+1) **do**

    1. Draw batch images $\{x_{(i)}^+\}_{i=1}^m$ from training set, where $(i)$ indicates a randomly selected index for sample $i$, and get samples $X_i^+ = x_{(i)} + \tau_{\text{data}}\epsilon_i$, where i.i.d. $\epsilon_i \sim \text{N}(0, I_D)$.

    2. Draw initial negative samples $\{Y_i^{(0)}\}_{i=1}^m$ from persistent image bank. Update $\{Y_i^{(0)}\}_{i=1}^m$ with the Langevin equation

$$Y_i^{(k)} = Y_i^{(k-1)} - \Delta\tau\nabla_{Y_\tau}f_{\theta_j}(Y_i^{\tau-1}) + \sqrt{2\Delta\tau}\epsilon_{i,k},$$

    where $\epsilon_{i,k} \sim \text{N}(0, I_D)$ i.i.d., for $K$ steps to obtain samples $\{X_i^-\}_{i=1}^m = \{Y_i^{(K)}\}_{i=1}^m$. Update persistent image bank with images $\{Y_i^{(K)}\}_{i=1}^m$.

    3. Update the weights by $\theta_{j+1} = \theta_j - g(\Delta\theta_j)$, where $g$ is the optimizer and

$$\Delta\theta_j = \frac{\partial}{\partial\theta}\left(\frac{1}{n}\sum_{i=1}^n f_{\theta_j}(X_i^+) - \frac{1}{m}\sum_{i=1}^m f_{\theta_j}(X_i^-)\right)$$

    is the ML gradient approximation.
**end for**

---

where $\beta_t \in (0, 1)$ is a variance schedule. After $T$ steps, $\boldsymbol{x}_T$ is nearly an isotropic Gaussian distribution. This reverse process is parameterized by a neural network and trained to de-noise a variable from the prior to match the real data distribution. Generating from a standard DDPM involves drawing samples from the prior, and then running the learned de-noising process to gradually remove noise and yield a final sample.

# B  PUREGEN Further Intuition

## B.1  Why EBM Langevin Dynamics Purify

The theoretical basis for eliminating adversarial signals using MCMC sampling is rooted in the established steady-state convergence characteristic of Markov chains. The Langevin update, as specified in Equation (4), converges to the distribution $p_\theta(x)$ learned from unlabeled data after an infinite number of Langevin steps. The memoryless nature of a steady-state sampler guarantees that after enough steps, all adversarial signals will be removed from an input sample image. Full mixing between the modes of an EBM will undermine the original natural image class features, making classification impossible [29]. Nijkamp et al. [2020] reveals that without proper tuning, EBM learning heavily gravitates towards *non-convergent ML* where short-run MCMC samples have a realistic appearance and long-run MCMC samples have unrealistic ones. In this work, we use image initialized *convergent learning*. $p_\theta(x)$ is described further by Algorithm 1 [48].

The metastable nature of EBM models exhibits characteristics that permit the removal of adversarial signals while maintaining the natural image's class and appearance [29]. Metastability guarantees that over a short number of steps, the EBM will sample in a local mode, before mixing between modes. Thus, it will sample from the initial class and not bring class features from other classes in its learned distribution. Consider, for instance, an image of a horse that has been subjected to an adversarial $\ell_\infty$ perturbation, intended to deceive a classifier into misidentifying it as a dog. The perturbation, constrained by the $\ell_\infty$-norm ball, is insufficient to shift the EBM's recognition of the image away from the horse category. Consequently, during the brief sampling process, the EBM actively replaces the adversarially induced 'dog' features with characteristics more typical of horses, as per its learned distribution resulting in an output image resembling a horse more closely than a dog. It is important

to note, however, that while the output image aligns more closely with the general characteristics of a horse, it does not precisely replicate the specific horse from the original, unperturbed image.

We use mid-run chains for our EBM defense to remove adversarial signals while maintaining image features needed for accurate classification. The steady-state convergence property ensures adversarial signals will eventually vanish, while metastable behaviors preserve features of the initial state. We can see PUREGEN-EBM sample purification examples in Fig. 5 below and how clean and poisoned sampled converge and poison perturbations are removed in the mid-run region (100-2000 steps for us).

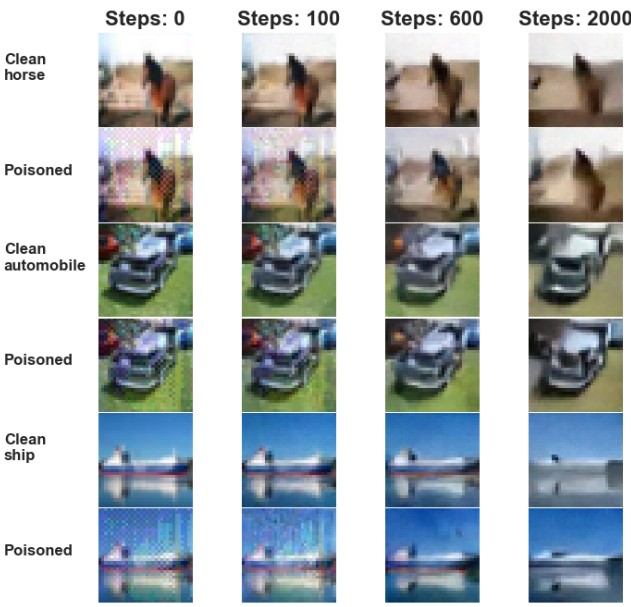

Figure 5: PUREGEN-EBM purification with various MCMC steps

Our experiments show that the mid-run trajectories (100-1000 MCMC steps) we use to preprocess the dataset $\mathcal{X}$ capitalize on these metastable properties by effectively purifying poisons while retaining high natural accuracy on $F_\phi(x)$ with no training modification needed. Intuitively, one can think of the MCMC process as a directed random walk toward a low-energy, more probable natural image version of the original image. The mid-range dynamics stay close to the original image, but in a lower energy manifold which removes the majority of poisoned perturbations.

## B.2 PUREGEN Additional L2 Dynamics Images

Additional L2 Dynamics specifically for Narcissus $\epsilon = 16$ are shown in Figures 6 and 7.

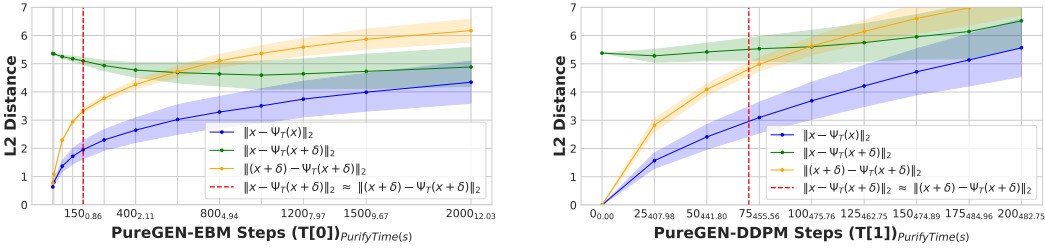

Figure 6: L2 Dynamics for Narcissus $\epsilon = 16$

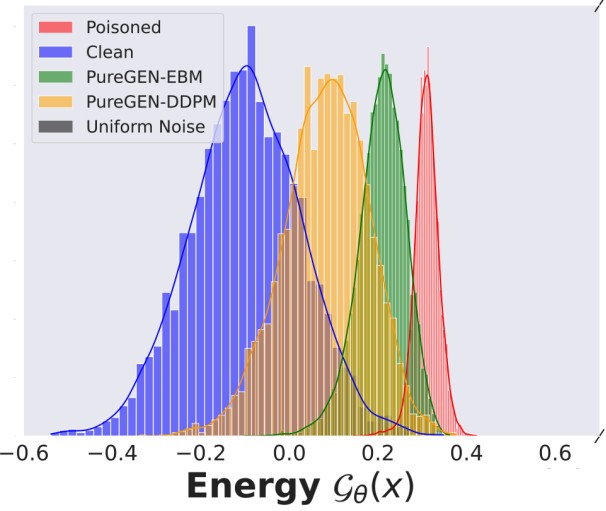

Figure 7: Energy distributions with Narcissus $\epsilon = 16$ poison.

## C    PUREGEN Extensions on Higher Power Attacks

We continue with the notation introduced in 3.3, where $\Psi_T(x)$ is the random variable of a fixed image $x$, and we define $T = (T_{\text{EBM}}, T_{\text{DDPM}}, T_{\text{Reps}}) \in \mathbb{R}^3$, where $T_{\text{EBM}}$ represents the number of EBM MCMC steps, $T_{\text{DDPM}}$ represents the number of diffusion steps, and $T_{\text{REPS}}$ represents the number of times these steps are repeated.

To incorporate EBM filtering, we order $\mathcal{D}$ by $\mathcal{G}_\theta(x)$ and partition the ordering based on $k$ into $\mathcal{D}_{max}^{(k)} \cup \mathcal{D}_{min}^{(1-k)}$, where $\mathcal{D}_{max}^{(k)}$ contains $k|\mathcal{D}|$ datapoints with the maximal energy (where $k = 1$ results in purifying everything and $k = 0$ is traditional training). Then, with some abuse of notation,

$$\Psi_{T,k}(\mathcal{D}) = \Psi_T(\mathcal{D}_{max}^{(k)}) \cup \mathcal{D}_{min}^{(1-k)} \tag{14}$$

To leverage the strengths of both PUREGEN-EBM and PUREGEN-DDPM, we propose PUREGEN combinations:

1. PUREGEN-NAIVE ($\Psi_{T|T_{EBM}>0,T_{DDPM}>0,T_{Reps}=1}$): Apply a fixed number of PUREGEN-EBM steps followed by PUREGEN-DDPM steps. While this approach does improve the purification results compared to using either method alone, it does not fully exploit the synergy between the two techniques.

2. PUREGEN-REPS ($\Psi_{T|T_{EBM}>0,T_{DDPM}>0,T_{Reps}>1}$): To better leverage the strengths of both methods, we propose a repetitive combination, where we alternate between a smaller number of PUREGEN-EBM and PUREGEN-DDPM steps for multiple iterations.

3. PUREGEN-FILT ($\Psi_{T,k|T_{EBM}\geq0,T_{DDPM}\geq0,0<k<1}$): In this combination, we first use PUREGEN-EBM to identify a percentage of the highest energy points in the dataset, which are more likely to be samples with poisoned perturbations as shown in Fig. 1. We then selectively apply PUREGEN-EBM or PUREGEN-DDPM purification to these high-energy points.

We note that methods 2 and 3 require extensive hyperparameter search with performance sweeps using the HLB model in App F, as there was little intuition for the amount of reps ($T_{Reps}$) or the filtering threshold ($k$) needed. Thus, we do not include these methods in our core results, but instead show the added performance gains on higher power poisons in Table 4, both in terms of increased perturbation size $\epsilon = 16$ and increased poison % (and both together). We note that 10% would mean the adversary has poisoned the entire class in CIFAR-10 with an NS trigger, and $\epsilon = 16$ is starting to approach visible perturbations, but both are still highly challenging scenarios worth considering for purification.

# D  Poison Sourcing and Experiment Implementation Details

Triggerless attacks GM and BP poison success refers to the number of single-image targets successfully flipped to a target class (with 50 or 100 target image scenarios) while the natural accuracy is averaged across all target image training runs. Triggered attack Narcissus poison success is measured as the number of non-class samples from the test dataset shifted to the trigger class when the trigger is applied, averaged across all 10 classes, while the natural accuracy is averaged across the 10 classes on the un-triggered test data. We include the worst-defended class poison success. The Poison Success Rate for a single experiment can be defined for triggerless $PSR_{notr}$ and triggered $PSR_{tr}$ poisons as:

$$PSR_{notr}(F, i) = \mathbb{1}_{F(x_i^\pi)=y_i^{\text{adv}}} \tag{15}$$

$$PSR_{tr}(F, y^\pi) = \frac{\sum_{(x,y)\in\mathcal{D}_{test}\setminus\mathcal{D}_{test}^\pi} \mathbb{1}_{F(x+\rho^\pi)=y^\pi}}{|\mathcal{D}_{test}\setminus\mathcal{D}_{test}^\pi|} \tag{16}$$

Note that all results except for Poison Success Rate for GM and BP attacks have a standard deviation, since those attacks are based on a single image class flip for a single classifiers training run. We do provide a subset with results for a single poison paradigm of BP and GM in App. E.2.3 where we used 3 different seeds for the training 3 classifiers for each of the 50 and 100 runs respectively to get a standard deviation, showing these results are low variance relative to the difference in results between baselines and our method. The compute required to collect such results across all scenarios would be extensive. .

## D.1  Poison Sourcing

### D.1.1  Bullseye Polytope

The Bullseye Polytope (BP) poisons are sourced from two distinct sets of authors. From the original authors of BP [9], we obtain poisons crafted specifically for a black-box scenario targeting ResNet18 and DenseNet121 architectures, and grey-box scenario for MobileNet (used in poison crafting). These poisons vary in the percentage of data poisoned, spanning 1%, 2%, 5% and 10% for the linear-transfer mode and a single 1% fine-tune mode for all models over a 500 image transfer dataset. Each of these scenarios has 50 datasets that specify a single target sample in the test-data. We also use a benchmark paper that provides a pre-trained white-box scenario on CIFAR-100 [50]. This dataset includes 100 target samples with strong poison success, but the undefended natural accuracy baseline is much lower.

### D.1.2  Gradient Matching

For GM, we use 100 publicly available datasets provided by [8]. Each dataset specifies a single target image corresponding to 500 poisoned images in a target class. The goal of GM is for the poisons to move the target image into the target class, without changing too much of the remaining test dataset using gradient alignment. Therefore, each individual dataset training gives us a single datapoint of whether the target was correctly moved into the poisoned target class and the attack success rate is across all 100 datasets provided.

### D.1.3  Narcissus

For Narcissus triggered attack, we use the same generating process as described in the Narcissus paper, we apply the poison with a slight change to more closely match with the baseline provided by [50]. We learn a patch with $\epsilon = 8/255$ on the entire 32-by-32 size of the image, per class, using the Narcissus generation method. We keep the number of poisoned samples comparable to GM for from-scratch experiment, where we apply the patch to 500 images (1% of the dataset) and test on the patched dataset without the multiplier. In the fine-tune scenarios, we vary the poison% over 1%, 2.5%, and 10%, by modifying either the number of poisoned images or the transfer dataset size (specifically 20/2000, 50/2000, 50/500 poison/train samples).

#### D.1.4 Neural Tangent Availability Attacks

For Neural Tangent Availability Attack, the full NTGA dataset (all samples poisoned) is sourced from the authors of the original NTGA attack paper [12]. Baseline defenses are pull from AVATAR [30].

### D.2 Training Parameters

We follow the training hyperparameters given by [18, 4, 9, 50] for GM, NS, BP Black/Gray-Box, and BP White-Box respectively as closely as we can, with moderate modifications to align poison scenarios. HyperlightBench training followed the original creators settings and we only substituted in a poisoned dataloader [46].

| Parameter | From Scratch | Transfer Linear | Transfer Fine-Tune |
|---|---|---|---|
| PUREGEN-EBM Steps ($T_{EBM}$) | 150 | 500 | 1000 |
| PUREGEN-DDPM Steps ($T_{DDPM}$) | 75 | 125 | 125 |
| PUREGEN-REPS ($T_{Reps}$) | 7 | - | - |
| PUREGEN-FILT (k) | 0.5 | - | - |
| Device Type | TPU-V3 | TPU-V3 | TPU-V3 |
| Weight Decay | 5e-4 | 5e-4 | 5e-4 |
| Batch Size | 128 | 64 | 128 |
| Augmentations | RandomCrop(32, padding=4) | None | None |
| Epochs | 200 or 80 | 40 | 60 |
| Optimizer | SGD(momentum=0.9) | SGD | Adam |
| Learning Rate | 0.1 | 0.1 | 0.0001 |
| Learning Rate Schedule (Multi-Step Decay) | 100, 150 - 200 epochs 30, 50, 70 - 80 epochs | 15, 25, 35 | 15, 30, 45 |
| Reinitialize Linear Layer | NA | True | True |

### D.3 Core Results Compute

Training compute for *core result only* which is in Table 1 on a TPU V3.

Table 6: Compute Hours TPU V3

| | From Scratch | | | | Transfer | | | Total |
|---|---|---|---|---|---|---|---|---|
| | Narcissus | Gradient Matching | NTGA | Fine Tune BP BlackBox | Fine Tune Narc | Linear BP BlackBox | Linear BP WhiteBox | |
| Train Time (Hours) | 1959.91 | 4155.74 | 283.75 | 73.82 | 45.89 | 548.15 | 70.76 | 7138.03 |

# E  Additional Results

## E.1  Extended Core Results

### E.1.1  Full Results Primary Experiments

Results on all primary poison scenarios with ResNet18 classifier including all EPIC versions (various subset sizes and selection frequency), FRIENDS versions (bernouilli or gaussian added noise trasnform), and all natural JPEG versions (compression ratios). Green highlight indicates a baseline defense that was selected for the main paper results table *chosen by the best poison defense performance that did not result in significant natural accuracy degradaion.* For both TinyImageNet and CINIC-10 from-scratch results, best performing baseline settings were used from respective poison scenarios in CIFAR-10 for compute reasons (so there are no additonal results and hence they are removed from this table).

**From Scratch**

**CIFAR-10 (ResNet-18)**

| | Gradient Matching-1% | | Narcissus-1% | | |
|---|---|---|---|---|---|
| | Poison Success (%) ↓ | Avg Nat Acc (%) ↑ | Avg Poison Success (%) ↓ | Avg Nat Acc (%) ↑ | Max Poison Success (%) ↓ |
| None | 44.00 | $94.84_{0.2}$ | $43.95_{33.6}$ | $94.89_{0.2}$ | 93.59 |
| EPIc-0.1 | 34.00 | $91.27_{0.4}$ | $30.18_{32.2}$ | $91.17_{0.2}$ | 81.50 |
| EPIc-0.2 | 21.00 | $88.04_{0.7}$ | $32.50_{33.5}$ | $86.89_{0.5}$ | 84.39 |
| EPIc-0.3 | 10.00 | $85.14_{1.2}$ | $27.31_{34.0}$ | $82.20_{1.1}$ | 84.71 |
| FRIENDS-B | 1.00 | $91.16_{0.4}$ | $8.32_{22.3}$ | $91.01_{0.4}$ | 71.76 |
| FRIENDS-G | **0.00** | $91.15_{0.4}$ | $9.49_{25.9}$ | $91.06_{0.2}$ | 83.03 |
| JPEG-25 | **0.00** | $90.00_{0.19}$ | $\mathbf{1.67}_{0.88}$ | $90.15_{0.26}$ | 3.38 |
| JPEG-50 | **0.00** | $91.70_{0.18}$ | $\mathbf{1.70}_{0.98}$ | $91.83_{0.20}$ | 3.83 |
| JPEG-75 | 2.00 | $92.73_{0.20}$ | $\mathbf{1.78}_{1.17}$ | $\mathbf{92.94}_{0.15}$ | 4.13 |
| JPEG-85 | 5.00 | $93.43_{0.16}$ | $5.76_{13.24}$ | $93.43_{0.20}$ | 43.36 |
| **PUREGEN-DDPM** | **0.00** | $90.93_{0.20}$ | $\mathbf{1.64}_{0.82}$ | $90.99_{0.22}$ | **2.83** |
| **PUREGEN-EBM** | 1.00 | $92.98_{0.2}$ | $\mathbf{1.39}_{0.8}$ | $\mathbf{92.92}_{0.2}$ | **2.50** |

**Transfer Learning (CIFAR-10, ResNet-18)**

| | Fine-Tune | | | | | Linear - Bullseye Polytope | | | |
|---|---|---|---|---|---|---|---|---|---|
| | Bullseye Polytope-10% | | Narcissus-10% | | | BlackBox-10% | | WhiteBox-1% | |
| | Poison Success (%) ↓ | Avg Nat Acc (%) ↑ | Avg Poison Success (%) ↓ | Avg Nat Acc (%) ↑ | Max Poison Success (%) ↓ | Poison Success (%) ↓ | Avg Nat Acc (%) ↑ | Poison Success (%) ↓ | Avg Nat Acc (%) ↑ |
| None | 46.00 | $89.84_{0.9}$ | $33.41_{33.9}$ | $90.14_{2.4}$ | 98.27 | 93.75 | $83.59_{2.4}$ | 98.00 | $70.09_{0.2}$ |
| EPIc-0.1 | 50.00 | $89.00_{1.8}$ | $32.40_{33.7}$ | $90.02_{2.2}$ | 98.95 | 91.67 | $83.48_{2.9}$ | 98.00 | $69.35_{0.3}$ |
| EPIc-0.2 | 42.00 | $81.95_{5.6}$ | $20.93_{27.1}$ | $88.58_{2.0}$ | 91.72 | 66.67 | $84.34_{3.8}$ | 91.00 | $64.79_{0.7}$ |
| EPIc-0.3 | 44.00 | $86.75_{6.3}$ | $28.01_{34.9}$ | $84.36_{6.3}$ | 99.91 | 66.67 | $83.23_{3.8}$ | 63.00 | $60.86_{1.5}$ |
| FRIENDS-B | 8.00 | $87.80_{1.1}$ | $3.34_{5.7}$ | $89.62_{0.5}$ | 19.48 | 35.42 | $84.97_{2.2}$ | 19.00 | $60.85_{0.6}$ |
| FRIENDS-G | 8.00 | $87.82_{1.2}$ | $3.04_{5.1}$ | $89.81_{0.5}$ | 17.32 | 33.33 | $85.18_{2.3}$ | 19.00 | $60.90_{0.6}$ |
| JPEG-25 | 0.00 | $88.93_{0.66}$ | $2.95_{3.71}$ | $87.63_{0.49}$ | 12.55 | **0.00** | $92.44_{0.47}$ | 8.0 | $50.42_{0.73}$ |
| JPEG-50 | **0.00** | $90.40_{0.44}$ | $3.51_{4.64}$ | $88.41_{0.58}$ | 15.76 | **0.00** | $86.03_{2.23}$ | 16.0 | $53.49_{0.54}$ |
| JPEG-75 | 4.00 | $90.11_{0.78}$ | $18.28_{25.83}$ | $89.12_{0.51}$ | 86.39 | 16.67 | $84.23_{1.76}$ | 36.0 | $56.02_{0.50}$ |
| JPEG-85 | 5.00 | $93.43_{0.16}$ | $25.19_{31.44}$ | $88.63_{1.59}$ | 94.41 | 54.17 | $83.61_{1.36}$ | 51.0 | $58.08_{0.54}$ |
| **PUREGEN-DDPM** | **0.00** | $\mathbf{91.53}_{0.15}$ | $\mathbf{1.88}_{1.12}$ | $90.69_{0.26}$ | **3.42** | **0.00** | $93.81_{0.08}$ | 9.0 | $54.53_{0.64}$ |
| **PUREGEN-EBM** | **0.00** | $87.52_{1.2}$ | $\mathbf{2.02}_{1.0}$ | $\mathbf{89.78}_{0.6}$ | **3.85** | **0.00** | $92.38_{0.3}$ | 6.00 | $\mathbf{64.98}_{0.3}$ |

### E.1.2 From Scratch 80 Epochs Experiments

Baseline FRIENDS [26] includes an 80-epoch from-scratch scenario to show poison defense on a faster training schedule. None of these results are included in the main paper, but we show again SoTA or near SoTA for PUREGEN against all baselines (and JPEG is again introduced as a baseline).

Table 7: From-Scratch 80-Epochs Results (ResNet-18, CIFAR-10)

| | **From Scratch (80 - Epochs)** | | | | |
|---|---|---|---|---|---|
| | Gradient Matching-1% | | Narcissus-1% | | |
| | Poison Success (%) ↓ | Avg Nat Acc (%) ↑ | Avg Poison Success (%) ↓ | Avg Nat Acc (%) ↑ | Max Poison Success (%) ↓ |
| None | 47.00 | $93.79_{0.2}$ | $32.51_{30.3}$ | $93.76_{0.2}$ | 79.43 |
| EPIc-0.1 | 27.00 | $90.87_{0.4}$ | $24.15_{30.1}$ | $90.92_{0.4}$ | 79.42 |
| EPIc-0.2 | 28.00 | $91.02_{0.4}$ | $23.75_{29.2}$ | $89.72_{0.3}$ | 74.28 |
| EPIc-0.3 | 44.00 | $92.46_{0.3}$ | $21.53_{28.8}$ | $88.05_{1.1}$ | 80.75 |
| FRIENDS-B | 2.00 | $90.07_{0.4}$ | $\mathbf{1.42}_{0.8}$ | $90.06_{0.3}$ | **2.77** |
| FRIENDS-G | 1.00 | $90.09_{0.4}$ | $\mathbf{1.37}_{0.9}$ | $90.01_{0.2}$ | 3.18 |
| JPEG-25 | 1.00 | $88.73_{0.24}$ | $\mathbf{1.66}_{0.92}$ | $90.01_{0.20}$ | 3.18 |
| JPEG-50 | 2.00 | $90.55_{0.23}$ | $\mathbf{1.76}_{1.07}$ | $90.56_{0.28}$ | 3.67 |
| JPEG-75 | **0.00** | $91.69_{0.23}$ | $\mathbf{1.68}_{1.14}$ | $91.67_{0.23}$ | 3.79 |
| JPEG-85 | 4.00 | $92.31_{0.23}$ | $\mathbf{1.87}_{1.41}$ | $\mathbf{92.42}_{0.16}$ | 5.13 |
| **PUREGEN-DDPM** | 1.00 | $89.82_{0.26}$ | $\mathbf{1.54}_{0.82}$ | $90.00_{0.13}$ | **2.52** |
| **PUREGEN-EBM** | 1.00 | $\mathbf{92.02}_{0.2}$ | $\mathbf{1.52}_{0.8}$ | $\mathbf{92.02}_{0.3}$ | 2.81 |

### E.1.3 Full Results for MobileNetV2 and DenseNet121

Table 8: MobileNetV2 Full Results

| | From Scratch | | | | | Transfer Learning | | | | |
| | Gradient Matching-1% | | Narcissus-1% | | | Fine-Tune Narcissus-10% | | | Linear BP BlackBox-10% | |
| | Poison Success (%)↓ | Avg Nat Acc (%)↑ | Avg Poison Success (%)↓ | Avg Nat Acc (%)↑ | Max Poison Success (%)↓ | Avg Poison Success (%)↓ | Avg Nat Acc (%)↑ | Max Poison Success (%)↓ | Poison Success (%)↓ | Avg Nat Acc (%)↑ |
|---|---|---|---|---|---|---|---|---|---|---|
| None | 20.00 | $93.86_{0.2}$ | $32.70_{24.5}$ | $93.92_{0.1}$ | 73.97 | $23.59_{23.2}$ | $88.30_{1.2}$ | 66.54 | 81.25 | $73.27_{1.0}$ |
| EPIC-0.1 | 37.50 | $91.28_{0.2}$ | $40.09_{27.1}$ | $91.15_{0.2}$ | 79.74 | $23.25_{22.8}$ | $88.35_{1.0}$ | 65.97 | 81.25 | $69.78_{2.0}$ |
| EPIC-0.2 | 19.00 | $91.24_{0.2}$ | $38.55_{27.5}$ | $87.65_{0.5}$ | 74.72 | $19.95_{19.2}$ | $87.67_{1.3}$ | 50.05 | 56.25 | $54.47_{5.6}$ |
| EPIC-0.3 | 9.78 | $87.80_{1.6}$ | $22.35_{23.9}$ | $78.16_{9.9}$ | 69.52 | $21.70_{28.1}$ | $78.17_{6.0}$ | 74.96 | 58.33 | $58.74_{9.0}$ |
| FRIENDS-B | 6.00 | $84.30_{2.7}$ | $\mathbf{2.00}_{1.3}$ | $88.82_{0.6}$ | 4.88 | $2.21_{1.5}$ | $83.05_{0.7}$ | **5.63** | 41.67 | $68.86_{1.5}$ |
| FRIENDS-G | 5.00 | $88.84_{0.4}$ | $\mathbf{2.05}_{1.7}$ | $88.93_{0.3}$ | 6.33 | $2.20_{1.4}$ | $83.04_{0.7}$ | 5.42 | 47.92 | $68.94_{1.5}$ |
| JPEG-25 | 1.00 | $85.18_{0.31}$ | $2.43_{1.16}$ | $85.00_{0.24}$ | 4.40 | $6.28_{9.05}$ | $80.00_{1.04}$ | 29.25 | 2.08 | $73.14_{0.71}$ |
| JPEG-50 | 1.00 | $86.82_{0.34}$ | $2.30_{1.20}$ | $86.60_{0.13}$ | 3.99 | $6.76_{10.23}$ | $83.70_{0.94}$ | 33.34 | 12.50 | $76.12_{1.75}$ |
| JPEG-75 | 1.00 | $88.08_{0.34}$ | $2.46_{1.42}$ | $87.88_{0.23}$ | 4.87 | $13.84_{18.74}$ | $84.67_{1.41}$ | 52.96 | 68.75 | $73.11_{1.53}$ |
| JPEG-85 | 2.00 | $88.83_{0.31}$ | $10.03_{16.93}$ | $88.61_{0.34}$ | 52.46 | $14.93_{18.42}$ | $85.46_{1.31}$ | 56.82 | 77.08 | $71.99_{1.07}$ |
| PUREGEN-EBM | 1.00 | $90.93_{0.2}$ | $\mathbf{1.64}_{0.8}$ | $91.75_{0.1}$ | **2.91** | $3.66_{5.4}$ | $84.18_{0.5}$ | 18.85 | **0.00** | $78.57_{1.4}$ |
| PUREGEN-DDPM | 1.00 | $86.79_{0.26}$ | $2.13_{1.02}$ | $86.91_{0.23}$ | **3.74** | $3.41_{4.82}$ | $86.92_{0.39}$ | 16.79 | **0.00** | $83.14_{0.20}$ |

Table 9: DenseNet121 Full Results

| | From Scratch | | | | | Transfer Learning | | | | |
| | Gradient Matching-1% | | Narcissus-1% | | | Fine-Tune Narcissus-10% | | | Linear BP BlackBox-10% | |
| | Poison Success (%)↓ | Avg Nat Acc (%)↑ | Avg Poison Success (%)↓ | Avg Nat Acc (%)↑ | Max Poison Success (%)↓ | Avg Poison Success (%)↓ | Avg Nat Acc (%)↑ | Max Poison Success (%)↓ | Poison Success (%)↓ | Avg Nat Acc (%)↑ |
|---|---|---|---|---|---|---|---|---|---|---|
| None | 14.00 | $95.30_{0.1}$ | $46.52_{32.2}$ | $95.33_{0.1}$ | 91.96 | $56.52_{38.6}$ | $87.03_{2.8}$ | 99.56 | 73.47 | $82.13_{1.6}$ |
| EPIC-0.1 | 14.00 | $93.0_{0.3}$ | $43.38_{32.0}$ | $93.07_{0.2}$ | 88.97 | $53.97_{39.0}$ | $87.04_{2.8}$ | 99.44 | 62.50 | $78.88_{2.1}$ |
| EPIC-0.2 | 7.00 | $90.67_{0.5}$ | $41.97_{33.2}$ | $90.23_{0.6}$ | 86.85 | $43.66_{36.5}$ | $85.97_{2.6}$ | 97.17 | 41.67 | $70.13_{5.2}$ |
| EPIC-0.3 | 4.00 | $88.3_{1.0}$ | $32.60_{29.4}$ | $85.12_{2.4}$ | 71.50 | $43.24_{43.0}$ | $72.76_{10.8}$ | 100.00 | 66.67 | $70.20_{10.1}$ |
| FRIENDS-B | 1.00 | $91.33_{0.4}$ | $8.60_{21.2}$ | $91.55_{0.3}$ | 68.57 | $5.34_{9.9}$ | $88.62_{0.8}$ | 33.42 | 60.42 | $80.22_{1.9}$ |
| FRIENDS-G | 1.00 | $91.33_{0.4}$ | $10.13_{25.2}$ | $91.32_{0.4}$ | 81.47 | $5.55_{10.4}$ | $88.75_{0.6}$ | 34.91 | 56.25 | $80.12_{1.8}$ |
| JPEG-25 | **0.00** | $90.09_{0.17}$ | $1.68_{1.10}$ | $90.15_{0.26}$ | 3.62 | $2.46_{2.92}$ | $83.82_{0.81}$ | 9.79 | 0.00 | $78.67_{1.60}$ |
| JPEG-50 | **0.00** | $91.94_{0.20}$ | $1.90_{1.54}$ | $92.03_{0.22}$ | 5.41 | $3.07_{2.69}$ | $85.92_{1.07}$ | 8.70 | 12.50 | $84.24_{1.39}$ |
| JPEG-75 | **0.00** | $93.08_{0.19}$ | $2.73_{3.12}$ | $93.16_{0.07}$ | 8.88 | $32.21_{35.67}$ | $87.19_{1.28}$ | 98.64 | 27.08 | $81.31_{1.34}$ |
| JPEG-85 | 7.00 | $93.85_{0.19}$ | $13.36_{28.10}$ | $93.77_{0.27}$ | 90.77 | $38.70_{37.98}$ | $86.25_{2.29}$ | 97.19 | 70.83 | $80.57_{1.40}$ |
| PUREGEN-EBM | **0.00** | $92.85_{0.2}$ | $\mathbf{1.42}_{0.7}$ | $93.48_{0.1}$ | **2.60** | $2.48_{1.9}$ | $88.75_{0.5}$ | **7.41** | **0.00** | $89.29_{0.9}$ |
| PUREGEN-DDPM | 3.00 | $91.09_{0.24}$ | $1.71_{0.94}$ | $90.94_{0.23}$ | 2.97 | $2.79_{2.51}$ | $88.21_{0.62}$ | 8.95 | 0.00 | $89.02_{0.15}$ |

### E.1.4 PUREGEN Combos on Increased Poison Power

Table 10: PUREGEN Combos with Narcissus Increased Poison % and $\epsilon$

| | Narcissus $\epsilon = 8$ 1% | | | Narcissus $\epsilon = 8$ 10% | | | Narcissus $\epsilon = 16$ 1% | | | Narcissus $\epsilon = 16$ 10% | | |
| | Avg Poison Success (%)↓ | Avg Nat Acc (%)↑ | Max Poison Success (%)↓ | Avg Poison Success (%)↓ | Avg Nat Acc (%)↑ | Max Poison Success (%)↓ | Avg Poison Success (%)↓ | Avg Nat Acc (%)↑ | Max Poison Success (%)↓ | Avg Poison Success (%)↓ | Avg Nat Acc (%)↑ | Max Poison Success (%)↓ |
|---|---|---|---|---|---|---|---|---|---|---|---|---|
| None | $40.33_{38.63}$ | $93.61_{0.12}$ | 90.26 | | $84.57_{0.60}$ | 99.97 | $83.63_{12.09}$ | $93.67_{0.11}$ | 97.36 | $99.35_{0.81}$ | $84.58_{0.63}$ | 99.97 |
| EPIC-0.1 | $36.39_{34.52}$ | $92.02_{0.63}$ | 87.34 | $98.63_{2.06}$ | $83.12_{0.49}$ | 99.98 | $74.81_{13.75}$ | $92.22_{0.33}$ | 94.26 | $98.81_{3.04}$ | $82.89_{0.59}$ | 99.96 |
| EPIC-0.2 | $29.75_{31.82}$ | $88.33_{0.37}$ | 82.41 | $96.65_{3.73}$ | $79.75_{0.94}$ | 99.78 | $73.11_{14.02}$ | $88.24_{0.59}$ | 91.51 | $98.54_{1.43}$ | $79.73_{0.93}$ | 99.57 |
| EPIC 03 | $28.53_{33.15}$ | $83.00_{1.87}$ | 93.68 | $93.54_{12.99}$ | $76.31_{1.54}$ | 99.99 | $56.16_{17.91}$ | $82.94_{1.53}$ | 81.59 | $97.40_{3.51}$ | $75.95_{1.59}$ | 99.97 |
| FRIENDS-G | $10.01_{26.92}$ | $91.18_{0.30}$ | 86.53 | $32.40_{43.26}$ | $84.95_{1.70}$ | 99.89 | $18.90_{20.93}$ | $91.15_{0.34}$ | 71.46 | $73.56_{20.71}$ | $82.82_{0.42}$ | 97.31 |
| FRIENDS-B | $7.89_{20.37}$ | $91.25_{0.26}$ | 65.78 | $30.87_{41.67}$ | $85.17_{1.70}$ | 100.00 | $18.06_{16.88}$ | $91.16_{0.29}$ | 54.17 | $71.28_{22.90}$ | $82.83_{0.43}$ | 99.25 |
| JPEG-25 | $\mathbf{1.82}_{0.98}$ | $87.76_{0.15}$ | 3.46 | $16.95_{9.72}$ | $84.66_{1.51}$ | 33.96 | $11.85_{12.60}$ | $87.72_{0.19}$ | 36.90 | $79.28_{8.05}$ | $79.87_{0.79}$ | 91.99 |
| JPEG-50 | $\mathbf{1.75}_{1.07}$ | $89.35_{0.27}$ | 3.71 | $31.14_{14.99}$ | $84.15_{1.63}$ | 50.97 | $21.67_{17.81}$ | $89.32_{0.19}$ | 57.58 | $90.48_{6.66}$ | $81.14_{0.86}$ | 100.00 |
| JPEG-75 | $\mathbf{1.66}_{0.92}$ | $90.70_{0.14}$ | 3.30 | $56.99_{27.60}$ | $84.07_{1.51}$ | 99.53 | $34.06_{21.16}$ | $90.77_{0.13}$ | 65.14 | $92.00_{6.90}$ | $82.20_{0.73}$ | 99.86 |
| JPEG-85 | $4.67_{9.82}$ | $91.74_{0.12}$ | 32.52 | $69.75_{25.47}$ | $83.72_{1.39}$ | 100.00 | $42.21_{26.43}$ | $91.66_{0.24}$ | 82.82 | $93.22_{6.90}$ | $82.90_{0.53}$ | 99.83 |
| PUREGEN-DDPM | $1.72_{0.92}$ | $89.61_{0.14}$ | 3.20 | $\mathbf{6.38}_{5.16}$ | $85.86_{0.46}$ | 16.29 | $\mathbf{5.21}_{3.35}$ | $86.16_{0.19}$ | 13.32 | $69.38_{16.73}$ | $83.58_{1.02}$ | 89.35 |
| PUREGEN-EBM | $\mathbf{1.59}_{0.86}$ | $91.41_{0.16}$ | 3.01 | $52.48_{23.29}$ | $86.14_{1.82}$ | 99.86 | $7.35_{4.46}$ | $85.61_{0.25}$ | 16.94 | $77.50_{7.01}$ | $78.79_{0.93}$ | 100.00 |
| PUREGEN-NAIVE $_{T=[150,75,1]}$ | $1.71_{0.89}$ | $88.84_{0.16}$ | 3.17 | $10.43_{8.58}$ | $88.20_{0.54}$ | 27.42 | $\mathbf{5.20}_{2.61}$ | $85.95_{0.23}$ | 9.80 | $63.01_{15.24}$ | $83.14_{0.90}$ | 87.17 |
| PUREGEN-REPS $_{T=[10,50,5]}$ | $1.73_{0.84}$ | $87.25_{0.18}$ | 3.25 | $\mathbf{3.75}_{2.28}$ | $85.56_{0.22}$ | **7.74** | $4.95_{2.48}$ | $85.79_{0.18}$ | 10.75 | $53.79_{17.14}$ | $83.92_{1.02}$ | **81.09** |
| PUREGEN-FILT $_{T=[0,125,1],k=0.5}$ | $1.73_{0.89}$ | $90.61_{0.16}$ | **2.88** | $6.47_{6.98}$ | $86.08_{2.00}$ | 18.81 | $5.74_{4.05}$ | $90.52_{0.18}$ | 16.08 | $69.13_{12.94}$ | $85.47_{1.45}$ | 87.66 |

## E.2 Additional Experiment Results

### E.2.1 Defense/EBM-Aware Narcissus Experiments

To further demonstrate the robustness of PUREGEN, we conduct an additional experiment simulating a scenario where an adversary is aware that PUREGEN is in use. Specifically, we incorporate the Energy-Based Model (EBM) within the Narcissus poison generation framework. Narcissus generates poisons by training a surrogate model, then refining the poison through SGD on frozen surrogate outputs (further details in [4]). For simplicity, we assume that the adversary possesses knowledge of the EBM defense mechanism but is unaware of the specific image instances ultimately used during training.

In this experiment, we include our pre-trained EBM as a preprocessing layer for the ResNet-based surrogate model, using 50 EBM steps——a choice that balances computational efficiency with reliable performance. When training the EBM-aware surrogate, we freeze the EBM to allow the classifier to train on a diverse set of stochastic EBM outputs. After a brief warmup phase, the entire surrogate

model is frozen, and we propagate gradients to optimize poison effectiveness. We test variations of when the surrogate and poison generation do or do not have EBM information, applied to 3 of the 10 CIFAR-10 classes. The results are recorded in Table 11.

Our results show that PUREGEN consistently defends against all new EBM-aware Narcissus poisons, demonstrating the robustness of our approach. Interestingly, we observe that when both the surrogate and poison-generating models include the EBM, the Narcissus method struggles to produce effective poisons, even in an undefended model context. Conversely, with a traditionally trained surrogate model but EBM augmentation in the poison generation process, the new poisons slightly improve efficacy on an undefended model. Overall, these findings reinforce the reliability of PUREGEN as a robust defense mechanism against adaptive poisoning attacks.

Table 11: Success Rates of EBM-Aware Narcissus Poisons Against PUREGEN Defense on classes 0,1,2. This table shows that even with the knowledge of the EBM, Narcissus is unable to generate effective poisons against PUREGEN. Furthermore, when the EBM is taken in the entire Narcissus process, Narcissus struggles to find a poison that works for undefended models.

| | Narcissus Baseline | | | Narcissus EBM Generation Model | | |
|---|---|---|---|---|---|---|
| | Avg Poison Success (%) $\downarrow$ | Avg Nat Acc (%) $\uparrow$ | Max Poison Success (%) $\downarrow$ | Avg Poison Success (%) $\downarrow$ | Avg Nat Acc (%) $\uparrow$ | Max Poison Success (%) $\downarrow$ |
| No Defense | $31.02_{49.32}$ | $93.64_{0.22}$ | 87.94 | $46.82_{37.92}$ | $93.36_{0.14}$ | 90.52 |
| **PUREGEN-DDPM** | $\mathbf{1.49}_{0.83}$ | $91.12_{0.10}$ | 2.36 | $\mathbf{1.06}_{0.53}$ | $91.10_{0.09}$ | 1.64 |
| **PUREGEN-EBM** | $\mathbf{1.57}_{0.82}$ | $91.48_{0.23}$ | 2.37 | $\mathbf{1.29}_{0.59}$ | $91.43_{0.15}$ | 1.83 |
| | Narcissus EBM Surrogate Model | | | Narcissus EBM Surrogate and Generation Model | | |
| | Avg Poison Success (%) $\downarrow$ | Avg Nat Acc (%) $\uparrow$ | Max Poison Success (%) $\downarrow$ | Avg Poison Success (%) $\downarrow$ | Avg Nat Acc (%) $\uparrow$ | Max Poison Success (%) $\downarrow$ |
| No Defense | $20.08_{32.55}$ | $93.57_{0.17}$ | 57.67 | $3.04_{2.46}$ | $93.65_{0.12}$ | 5.74 |
| **PUREGEN-DDPM** | $\mathbf{1.46}_{1.06}$ | $91.04_{0.17}$ | 2.64 | $\mathbf{1.42}_{1.03}$ | $91.12_{0.11}$ | 2.60 |
| **PUREGEN-EBM** | $\mathbf{1.56}_{1.03}$ | $91.36_{0.23}$ | 2.73 | $\mathbf{1.59}_{0.82}$ | $91.29_{0.15}$ | 2.50 |

### E.2.2  Pre-Trained Public DDPM Comparison

We include results using two pre-trained diffusion models from HuggingFace [51, 52]. The results show that these models can achieve defense performance similar to that of some of our POOD in-house trained models. The table below includes 4 baseline PUREGEN models and the two Hugging Face models trained on butterflies and anime datasets, showing both are comparable for poison defense and natural accuracy to some POOD datasets in performance. **These results how that, for PUREGEN-DDPM pre-trained models could be adequate**, but come with the risks of using a model with unknown data security. We reiterate our primary contribution for PUREGEN-DDPM was in reducing training and improving performance for a given architecture and dataset if one needs to train a diffusion model and if purification is the known use-case.

Table 12: Two pre-trained diffusion models from HuggingFace, showing similar results to our POOD DDPM results on Narcissus From-Scratch attack [51, 52].

| DDPM Model | Poison Success (%) | Nat Acc (%) | Max Poison (%) |
|---|---|---|---|
| PUREGEN-EBM CINIC-10_IN | **1.39 ± 0.80** | **92.92 ± 0.20** | **2.50** |
| PUREGEN-DDPM CINIC-10_IN | 1.64 ± 0.82 | 90.99 ± 0.22 | 3.83 |
| PUREGEN-DDPM Food-101 | 1.71 ± 0.74 | 88.35 ± 0.21 | 2.72 |
| PUREGEN-DDPM Office-Home | 1.80 ± 0.83 | 87.32 ± 0.22 | 3.16 |
| HuggingFace Butterflies [51] | 1.65 ± 0.83 | 87.79 ± 0.18 | 3.01 |
| HuggingFace Anime [52] | 1.47 ± 0.75 | 90.91 ± 0.13 | 2.95 |

### E.2.3 GM and BP Poison Success Standard Deviation

Table 13: Core results poison success for one GM and one BP scenario where we compute poison success across 3 different seeds to show the relatively low variance of these results where our method is still SoTA.

| Poison Scenario | Poison Success (%) | | | | | |
| --- | --- | --- | --- | --- | --- | --- |
| | Baseline | EPIC | FRIENDs | JPEG | PUREGEN-EBM | PUREGEN-DDPM |
| From-Scratch, GM-1%, CIFAR-10 (ResNet-18) | $44.44_{2.17}$ | $10.58_{4.17}$ | $\mathbf{0.33}_{0.47}$ | $\mathbf{0.67}_{0.47}$ | $1.00_{0.00}$ | $\mathbf{0.00}_{0.00}$ |
| Fine-Tune, BP-10%, CIFAR-10 (ResNet-18) | $48.63_{6.18}$ | $44.67_{5.73}$ | $7.19_{4.8}$ | $\mathbf{0.67}_{0.94}$ | $\mathbf{0.0}_{0.0}$ | $\mathbf{0.67}_{0.94}$ |

# F  PureGen Extensions T Sweeps

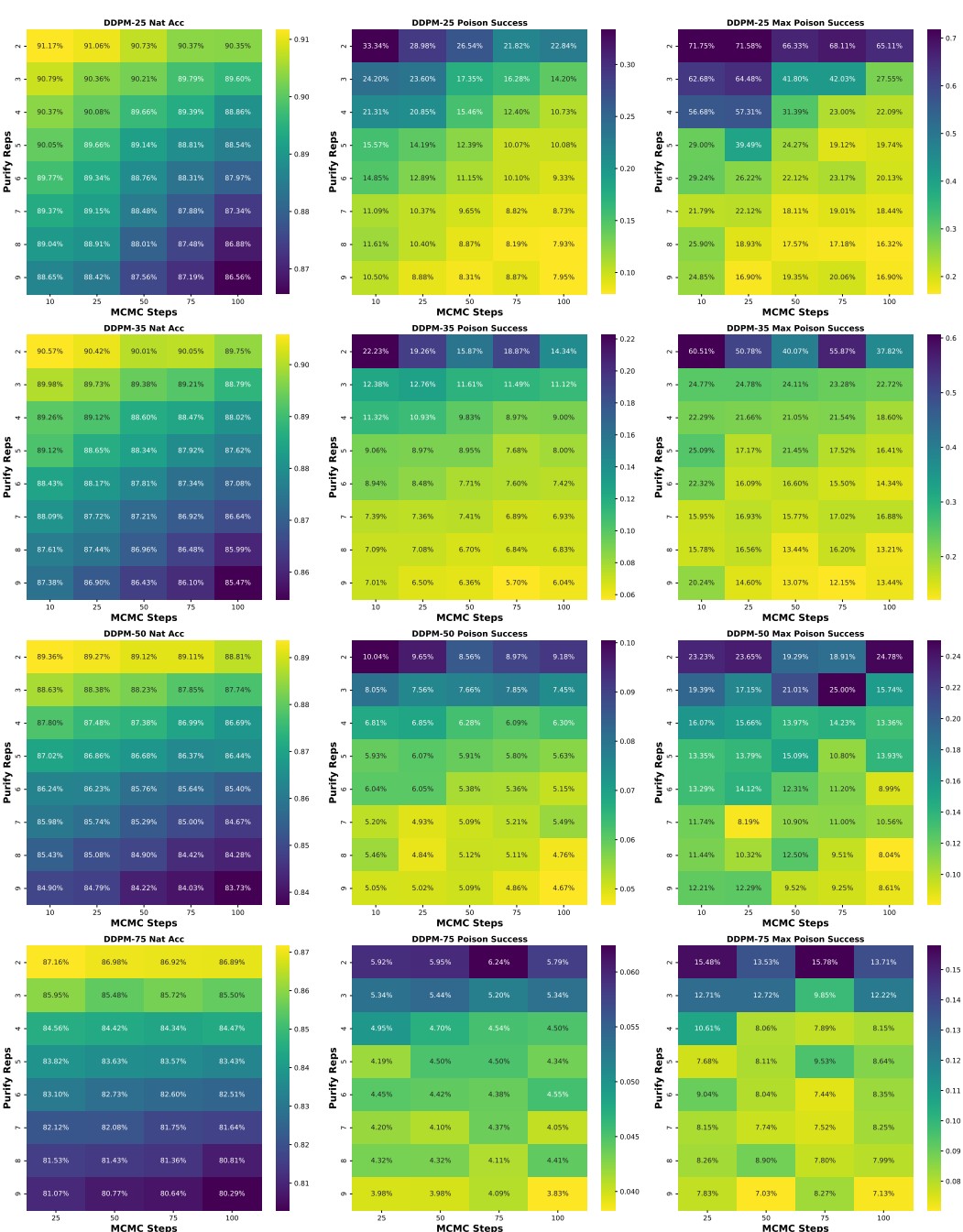

Figure 8: PureGen-Reps Sweeps with HLB Model on Narcissus

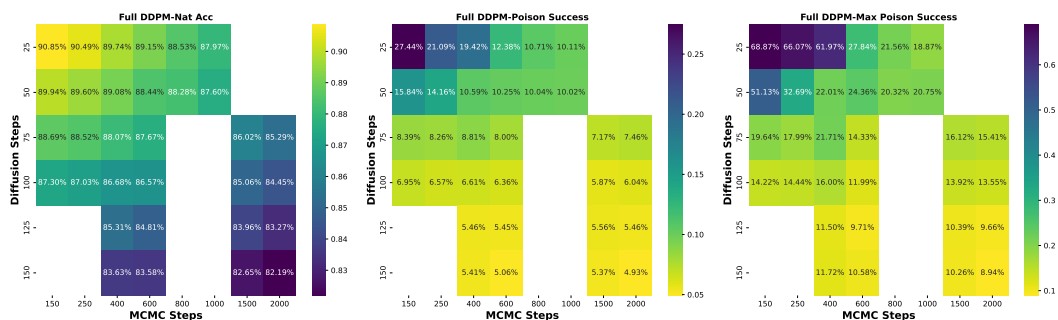

Figure 9: PUREGEN-NAIVE Sweeps with HLB Model on Narcissus

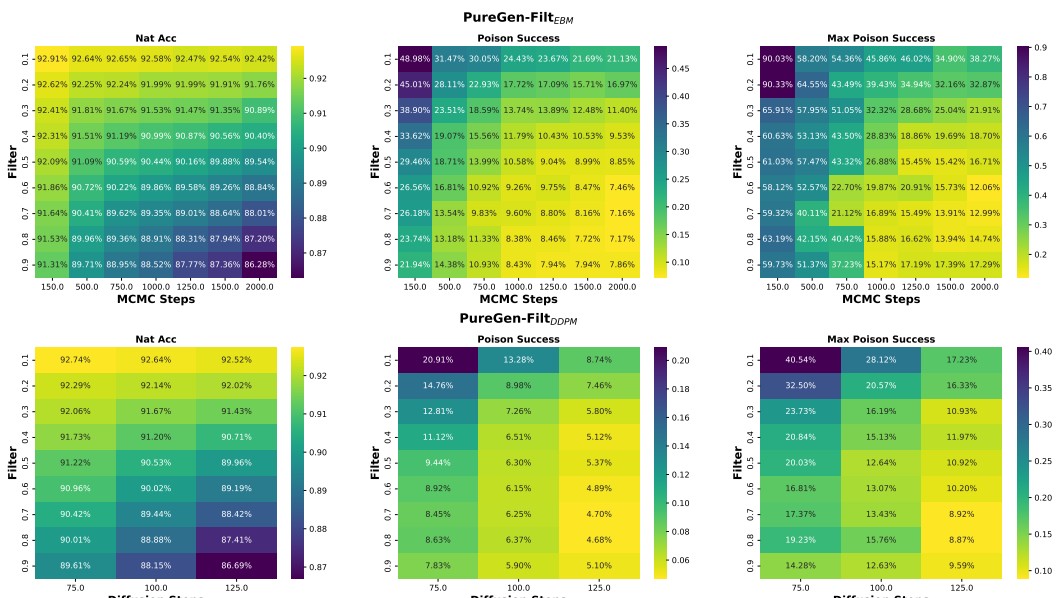

Figure 10: PUREGEN-FILT Sweeps with HLB Model on Narcissus

# G Interpreting PUREGEN Results

## G.1 Model Interpretability

Using the Captum interpretability library, in Figure 11, we compare a clean model with clean data to various defense techniques on a sample image poisoned with the NS Class 5 trigger $\rho$ [53]. Only the clean model and the model that uses PUREGEN-EBM correctly classify the sample as a horse, and the regions most important to prediction, via occlusion analysis, most resemble the shape of a horse in the clean and PUREGEN-EBM images. Integrated Gradient plots show how PUREGEN-EBM actually enhances interpretability of relevant features in the gradient space for prediction compared to even the clean NN.

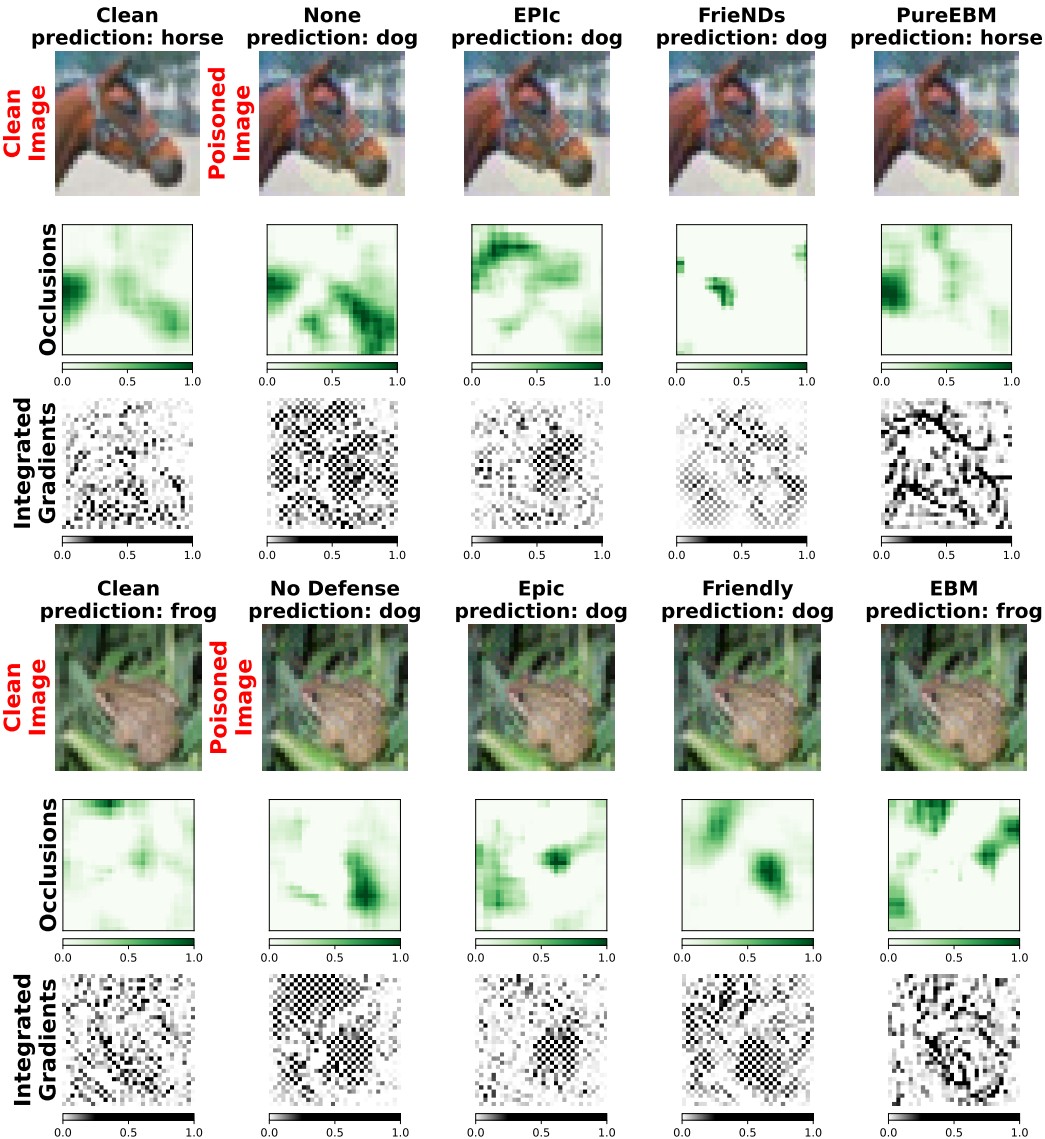

Figure 11: Defense Interpretability: Model using PUREGEN-EBM focuses on the outline of the horse in the occlusions analysis and to a higher degree on the primary features in the gradient space than even the clean model on clean data.

## G.2 Differences between PUREGEN-EBM and PUREGEN-DDPM

In this section we visualize a Narcissus $\epsilon = 64$ trigger patch to better see the PUREGEN-EBM and PUREGEN-DDPM samples on a visible perturbation. In Figure 12 we see again how the EBM struggles to purify larger perturbations but better preserves the image content, while DDPM can degrade such perturbations better at the cost of degrading the image content as well.

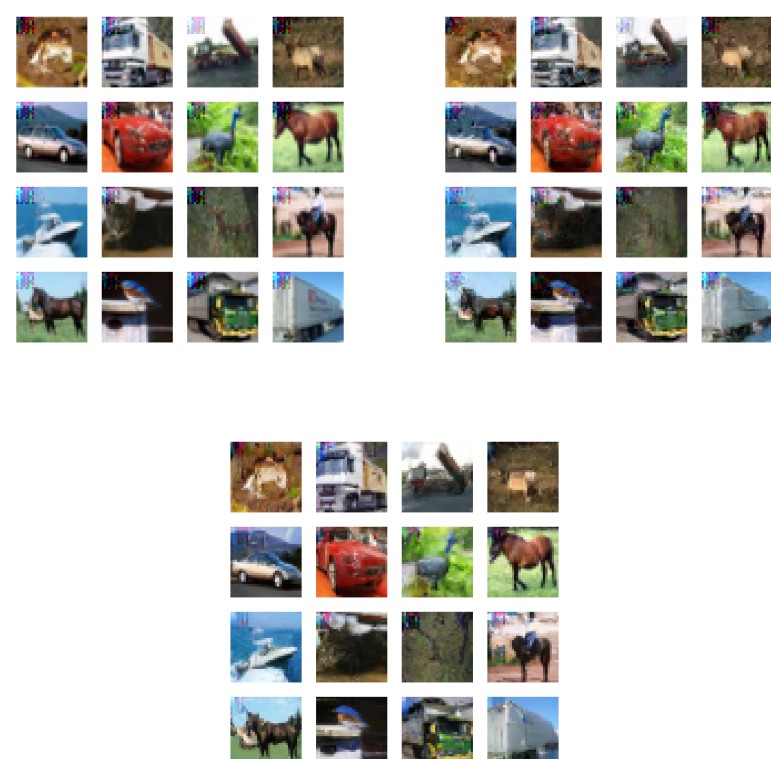

Figure 12: Narcissus $\epsilon = 64$ trigger patch purification samples **Top Left**: Original Poisoned. **Top Right**: PUREGEN-EBM 500 Steps. **Top Left**: PUREGEN-DDPM 75 Steps.

