# OpenReview forum: "PureGen: Universal Data Purification for Train-Time Poison Defense via Generative Model Dynamics"
_NeurIPS.cc/2024/Conference — NeurIPS 2024 poster_

### Official Review · Reviewer_K3RB · 2024-06-12

**Soundness:** 3
**Presentation:** 3
**Contribution:** 3
**Rating:** 5
**Confidence:** 4

**Summary:**

The paper presents a novel approach to securing model's performance (robust and natural accuracy) against train-time data poisoning attacks by introducing a set of data purification transformations during training, specifically employing Energy-Based Models (EBM) and Denoising Diffusion Probabilistic Models (DDPM). The effectiveness of the proposed method is demonstrated through evaluations on CIFAR-10, Tiny-ImageNet, and CINIC-10 datasets.

**Strengths:**

1. The paper is generally well-written and structured. However, certain sections could benefit from additional clarity and elaboration to enhance the overall presentation quality.
2. The elucidation of the L2 Distance in Section 3.4 is commendable, offering a lucid explanation of the method's underlying mechanism for aligning poisoned data with the benign data distribution.
3. The exploration of different variants of the proposed method in Section 4.4 is a valuable addition to the paper.
4. The method achieves state-of-the-art performance in the conducted experiments, with a comprehensive exploration of its capabilities across various scenarios.

**Weaknesses:**

1. Section 3.1 could be improved by providing a more in-depth discussion on the application of EBM models within the context of the proposed method, rather than focusing predominantly on foundational concepts.
2. The relevance of Section 3.3 is questionable, as the equations introduced (e.g., Eq. 6) do not appear to be integral to the subsequent discussion.
3. The paper may benefit from a clearer articulation of its novelty within the field.

**Questions:**

1. The application of EBM and DDPM for adversarial purification is gaining traction, particularly in defense against inference attacks [1]. Is this the first application of these models for training-time data poisoning defense? How does this work distinguish itself within the existing body of literature regarding its novelty?

2. The claim on Page 2, Line 45, that purification models require training with a POOD dataset may not present a significant challenge. Pre-trained DDPM models are readily available in open-source repositories and can be employed for data purification, as demonstrated in [1]. Could the authors conduct a comparative analysis between the performance of publicly available DDPM models and those trained in-house?


[1] "(Certified!!) Adversarial Robustness for Free!." Carlini, Nicholas, et al. The Eleventh International Conference on Learning Representations. 2022.

**Limitations:**

See weakness and questions.

---

> ### Author Rebuttal · Authors · 2024-08-07
>
> We thank the reviewer for their time and directly respond to the stated questions and weaknesses.
>
> ---
>
> ### Weaknesses
>
> 1. *The reviewer suggests that Section 3.1 could be improved by providing a more in-depth discussion on the application of EBM models within the context of the proposed method.*
>
> We appreciate this feedback and are open to modifying Section 3.1. As this is the first application of an EBM in the poison setting, we believe it was useful background information. That being said, we can condense the fundamental concepts more in the camera-ready version and move more details to the appendix.
>
> 2. *The reviewer questions the relevance of Section 3.3, as the equations introduced do not appear integral to the subsequent discussion.*
>
> While these sections provide a theoretical basis for a stochastic transformation defense, we can further condense this section down and move some details to the appendix. We will revise the paper to ensure a clearer connection between these theoretical concepts and their practical application in our method.
>
> 3. *The reviewer states that the paper may benefit from a clearer articulation of its novelty within the field.*
>
> We agree and have articulated the novelty in more detail in the general rebuttal **Novelty of Work** section. We will similarly update the camera-ready version of the paper.
>
> ---
>
> ### Questions
>
> 1. *Is this the first application of EBM and DDPM for training-time data poisoning defense? How does this work distinguish itself within the existing body of literature regarding its novelty?*
>
> Yes, this is the first application of EBM and DDPM models for training-time data poisoning defense to the best of our knowledge. We address the main novelty points of our paper in our general rebuttal **Novelty of Work** section and clarify the difference between train-time and test-time poisons.
>
> 2. *The claim on Page 2, Line 45, that purification models require training with a POOD dataset may not present a significant challenge. Could the authors conduct a comparative analysis between the performance of publicly available DDPM models and those trained in-house?*
>
> We have included an additional experiment using two pre-trained diffusion models from HuggingFace. The results show that these models can achieve defense performance similar to that of some of our POOD in-house trained models. The table below includes 4 baseline PureGen models and the two Hugging Face models trained on butterflies and anime datasets [5,6], showing both are comparable for poison defense and natural accuracy to some POOD datasets in performance. We will include these results and make clear that when available, a pre-trained diffusion model is quite capable of providing poison defense. Our primary insight was in reducing training and improving performance for a given architecture and dataset if one needs to train a diffusion model and if purification is the known use-case.
>
>
> |    | Model                  | Poison Success (%) |  Nat Acc (%) | Max Poison (%) |
> |---:|:-----------------------|:------------------:|:------------:|:--------------:|
> |  0 |PureGen-EBM CINIC-10_IN |    1.39 ± 0.80     | 92.92 ± 0.20 |      2.50     |
> |  1 |PureGen-DDPM CINIC-10_IN|    1.64 ± 0.82     | 90.99 ± 0.22 |      3.83      |
> |  2 |PureGen-DDPM Food-101   |    1.71 ± 0.74     | 88.35 ± 0.21 |      2.72      |
> |  3 |PureGen-DDPM Office-Home|    1.80 ± 0.83     | 87.32 ± 0.22 |      3.16      |
> |  4 |HuggingFace Butterflies |    1.65 ± 0.83     | 87.79 ± 0.18 |      3.01      |
> |  5 |HuggingFace Anime       |    1.47 ± 0.75     | 90.91 ± 0.13 |      2.95      |
>
>
> ---
>
> We hope these responses clarify any stated weaknesses and answer the reviewer’s questions. References are in general rebuttal. We look forward to any additional discussion.

---

> > ### Comment · Reviewer_K3RB · 2024-08-09
> >
> > Dear Authors,
> >
> > Thank you for the efforts you have made to address the concerns raised in my previous review. Upon careful consideration of your response, I would like to offer the following feedback:
> >
> >  - **Concern Regarding Novelty:** The concept of adversarial purification is indeed established within the literature. While its application during the training phase is an interesting direction, I am not entirely convinced that this alone constitutes a substantial contribution or presents a significant advancement to warrant a recommendation for acceptance in its current form.
> >
> >  - **Concern Regarding Performance:** The effectiveness of pre-trained models, as demonstrated in your work, raises important questions. Specifically, it prompts an inquiry into whether the natural image distribution encompasses the test dataset used in your evaluation.
> >
> > In light of these concerns, I believe that the manuscript would benefit from a major revision, particularly in the following area:
> >
> > Rather than focusing solely on the application of adversarial purification during the training phase as the primary novelty, I encourage you to delve deeper into the effects of data distribution. A thorough analysis and discussion on how data distribution influences the outcomes could significantly enhance the paper's contribution and elevate its importance within the field.
> >
> > I hope that these suggestions will assist you in refining your manuscript. Regrettably, I must maintain my current score, as the contribution does not yet meet the threshold for acceptance.
> >
> > Warm regards,

---

> > > ### Author Response · Authors · 2024-08-09
> > >
> > > Dear Reviewer,
> > >
> > > Thank you for responding to our rebuttal, and we are glad to respond to the feedback and concerns:
> > >
> > > * **Concern Regarding Novelty**: We disagree with this characterization since **our novel contributions include extensive distributional analysis and impact of poisoning on defense generative models (Section 4.4), combinations of generative models and energy-based filtering (Section 4.5), extensive analysis on purification and poisons in the energy space (Section 3.4), and the decreased DDPM training pipeline.** Train-time poisons have unique enough considerations and have been separated in the literature for a few years. Considering the growth of poisons and defenses within train-time poison literature alone since then, the fact that we comprehensively address SoTA, and the added novelties above, we believe the paper is a strong contribution to the field in its current form.
> > >
> > > * **Concern Regarding Performance and Data Distribution Analysis**: This experiment was done as per the reviewer's request, and these were the pre-trained models with the right resolution we could find. *All models in the paper have clearly defined train distributions that do not contain the test dataset (unless explicitly called out as a baseline as in the POOD analysis).* ***Further, Section 4.4 is entirely focused on the impact of data distribution on purification performance for both EBMs and DDPMs.*** We believe we thoroughly explore this topic on data distribution impact on purification (far beyond any paper that uses generative models in any adversarial setting). We hope you can further clarify what additional analysis would look like.
> > >
> > > We thank you again for the discussion and for giving us the added feedback. We believe your concerns are adequately addressed in the paper and in our clarifications. We hope you can reconsider your assessment with these comments.
> > >
> > > Kindly,
> > > Authors

---

### Official Review · Reviewer_n59K · 2024-06-26

**Soundness:** 3
**Presentation:** 3
**Contribution:** 2
**Rating:** 4
**Confidence:** 4

**Summary:**

This paper studies the generative purification methods, i.e., EBM and DDPM-based purifications, as defenses against a set of data poisoning attacks.

**Strengths:**

1. They suggest that a proper range of implementation steps in the EBM and DDPM-based purification methods matters for defense performance.

2. They conduct comprehensive experiments to evaluate the effectiveness of PureGen methods.
Experimental results show that proposed defenses perform better than the baselines in the paper.


3. They study some considerations in practical scenarios including (a) training distribution shift in the generative model training; (b) training the generative model on poisoned data; (c) network architecture transferability

4. They explore some PureGen variants.

**Weaknesses:**

1. The idea of employing generative methods to purify imperceptive noises has been explored, i.e. EBM [1] and DDPM [2].
As for training-time poisoning attacks, [3, 4] propose to use diffusion process to defend against availability attacks.
Though this paper studies different data poisoning attacks such as Bullseye Polytope, Gradient Matching, and Narcissus, I don't think there is any essential difference in the defense mechanism.

2. Since the technical modification in DDPM lies in the truncation of training steps, it's beneficial to move Figure 4 to the main paper which illustrates how the the selection of steps influences the defense performance.




[1] Yoon J, Hwang S J, Lee J. Adversarial purification with score-based generative models

[2] Nie W, Guo B, Huang Y, et al. Diffusion Models for Adversarial Purification

[3] Jiang W, Diao Y, Wang H, et al. Unlearnable examples give a false sense of security: Piercing through unexploitable data with learnable examples

[4] Dolatabadi H M, Erfani S, Leckie C. The devil’s advocate: Shattering the illusion of unexploitable data using diffusion models

**Questions:**

1. Can you provide more intuition about why sacrificing generative capabilities can improve poison defense?

2. Can AVATAR[1] work as a defense baseline against Bullseye Polytope, Gradient Matching, and Narcissus?

3. In Table 1, why are there no gradient matching results on CINIC-10 and no Narcissus results on Tiny-ImageNet?

4. Why do some Poison Success cells show no standard deviations in Table 1,2?
And is the bold font abused in these tables?

[1] Dolatabadi H M, Erfani S, Leckie C. The devil’s advocate: Shattering the illusion of unexploitable data using diffusion models

**Limitations:**

See weaknesses and questions.

---

> ### Author Rebuttal · Authors · 2024-08-07
>
> We thank the reviewer for their time and directly respond to the stated questions and weaknesses.
>
> ---
>
> ### Weaknesses
>
> 1. *The idea of employing generative methods to purify imperceptible noises has been explored before and the defense mechanism is not different from other attack paradigms such as the availability/unlearnable attack.*
>
> In the general rebuttal (**Train-Time vs. Inference-Time Attacks**), we detail the differences between train and inference-time attacks and availability attacks. Our initial EBM work is based on the 2020 paper Stochastic Security [1]. We adapted a similar setup in PureGen-EBM for the poison setting, showing it defends against poisons much better than SoTA. We found poisons are separable as high-energy, which is specific to train-time poison settings. We scaled up the EBM and optimized the number of steps. Given the popularity of diffusion models related to EBMs via Langevin sampling, we found they also reach near SoTA performance.
>
> Unlearnable examples (availability attacks) are recent, and so is the Devil's Advocate paper [4]. AVATAR is related to diffusion-style purification, which is why we compared PureGen-DDPM on availability examples in Table 2. We see that out-of-the-box PureGen performs nearly as well as AVATAR. Using AVATAR would be "cheating" since it has seen clean versions of poisoned images as it was designed for inference-time attacks. Additionally, the augmentations used in their method (Cut-Mix, Cut-Out, etc.) are not used in our method to isolate PureGen's impact. Unlearnability and clean-label train-time poisons pose different problems; unlearnability produces detectable results counter to neural network goals, while poisoning is harder to detect post-training, posing significant security risks. Our technique provides universal defense against the strongest poisoning techniques.
>
> Finally, we conducted numerous additional experiments for intuition, exploring POOD distribution shifts, poisoning generative model training data, and combining EBM+DDPM and EBM filtering, pushing the boundaries of generative model defense work. We believe these differences and extensive experiments make PureGen a valuable contribution to the community.
>
> 2. *The reviewer suggests moving Figure 4 to the main paper to illustrate how the selection of steps influences the defense performance.*
>
> We appreciate this suggestion and will move Figure 4 to the main paper for the camera-ready version to clarify the justification for our pipeline modification.
>
> ---
>
> ### Questions
> 1. *Can you provide more intuition about why sacrificing generative capabilities can improve poison defense?*
>
> The empirical results in Figure 4 support that for a given architecture, dataset, and training pipeline, sacrificing some generative capabilities improves poison defense. We utilize Langevin dynamics as a "restoration" of the conditional information in the corrupted image, akin to EBM dynamics moving a sample to a lower-energy, more realistic image. Since the model is not trained to generate from the prior, no model capacity is needed for this initial high-energy, random noise sample. All model capacity is dedicated to restoring the corrupted image, which retains significant low-energy information, similar to conditional generative diffusion processes like in-fill or super-resolution, rather than standard unconditional DDPMs. We will include more discussion around this in a camera-ready version.
>
> 2. *Can AVATAR work as a defense baseline against Bullseye Polytope, Gradient Matching, and Narcissus?*
>
> AVATAR is primarily designed for unlearnable examples, which differ from train-time poisoning attacks. The released AVATAR checkpoints are trained on CIFAR-10, making it unfair to use them for purifying unpoisoned images as they have memorized the clean images. However, diffusion methods trained on the correct subsets or POODs likely work, as shown by our results with pre-trained diffusion models in the general rebuttal (**Usage of Pre-Trained Diffusion Models**). While PureGen-DDPM training offers computational advantages and some performance improvements, it is not necessary to achieve reasonable defense performance when pre-trained models are available.
>
> 3. *Why are there no gradient matching results on CINIC-10 and no Narcissus results on Tiny-ImageNet?*
>
> We focused our experiments on the most relevant and challenging scenarios for each dataset, leveraging available poisons from crafting papers. Crafting new poisons is computationally expensive, highly hyper-parameter dependent, and not typically done by defense paper authors. We will clarify this in the appendix. We crafted “new” poisons for analysis in the “defense-aware Narcissus poison” in the general rebuttals, going beyond the scope of typical defense papers.
>
> 4. *Why do some Poison Success cells show no standard deviations in Table 1,2? And is the bold font abused in these tables?*
>
> Standard deviations are not shown for un-triggered poison scenarios (Gradient Matching and Bullseye Polytope) for Poison Success. For these experiments, the poison success is based on training 100 classifiers (as each classifier has a single target image for which it is poisoned or not). It would be possible to obtain a standard deviation using different initialization seeds, such repetitions are quite costly (core table results alone took over 7k TPU V3 hours). We can obtain these results for a camera-ready version.
>
> We use bold fonts to highlight the highest performing method for each category within reason (highest natural accuracy might be ignored if there is poor poison defense). The exception is Table 3 where we try to highlight all the poisoned model scenarios that would still be SoTA. We will modify Table 3 to use bold font for the highest performing methods as well.
>
> ---
>
> We hope these responses clarify any weaknesses and answer the reviewer’s questions. References are in the general rebuttal. We look forward to any additional discussion.

---

> ### Comment · Reviewer_n59K · 2024-08-11
>
> Thanks for your responses and clarifications; they have indeed addressed some of my concerns.
>
> However, I will maintain my score due to the remaining major concerns:
>
> - There is no fundamental difference between training-time purification and test-time purification. Before retraining, the poisoned samples have already been purified and fixed. In other words, purification is not truly integrated into the training process; purification and retraining are completely separate.
> I am not convinced by the story which applies existing diffusion purification to a new perturbation type and makes it the first work addressing this problem.
>
> - Main comparisons in the paper, including Tables 1 and 2, adopt no diffusion purification methods as baselines. However, additional results in rebuttal show that pre-trained diffusion models also work in purifying poisoning attacks. Considering the authors' claim that their methods provide SOTA defense,  I have concerns about whether the advantages of specially designed techniques are still solid under more comprehensive comparisons, especially with more pre-trained diffusion models.
>
> I indeed appreciate the studies regarding practical scenarios, as I stated in Strength 3, 4.
> Therefore, I agree with other review's suggestions that it might be better to shift the focus of this paper.

---

> > ### Author Response · Authors · 2024-08-11
> >
> > Respectfully, we feel the reviewer has a miss understanding of the point of the paper. In this paper we set-out to show users should purify their datasets with an EBM or diffusion model to get SoTA defense against training poisons with little degradation in natural accuracy. We consider how diffusion models that have been trained on out of distribution data perform in the purification task. We conclude that; 1) EBMs and diffusion models perform SoTA for defense while retraining nat acc, 2) OOD EBM/diffusion works so people should be purifying all image data with whatever EBM/diffusion model they have access to.
> >
> > Now we will go through the latest concerns of the reviewer.
> >
> > The reviewer says:
> >
> > "Before retraining, the poisoned samples have already been purified and fixed. In other words, purification is not truly integrated into the training process; purification and retraining are completely separate. "
> > - That is the entire point. Other defense methods that prevent training poisons slow down training and or reduce performance. Our method protects against training poisons by purifying the entire dataset. We compare to two other SoTA training defense methods; EPIC and FRIENDs. EPIC does not adjust the poisoned training images but instead rejects some, FRIENDS uses the classifier's state during training to calculate a perturbation to the image. Both of these methods require modification to the training loop and FRINDS is very computationally expensive.
> > - Our method as we stated thought the paper and should not be a shock to the reviewer at this point in the review cycle "fixes" or "purifies" the entire dataset before training begins. The class of poisons we are protecting against are named train time poisons. The defense does not require one to augment to training pipeline to be considered a defense.
> >
> > "I am not convinced by the story which applies existing diffusion purification to a new perturbation type and makes it the first work addressing this problem. "
> > - What does the reviewer need to be convinced of? We show that the perturbations intact defend, we show that when we are OOD we might degrade the natural accuracy and increase poison success but we simply share the empirical findings... If you do not believe us feel free to use our posed code to verify.
> >
> > "Main comparisons in the paper, including Tables 1 and 2, adopt no diffusion purification methods as baselines. However, additional results in rebuttal show that pre-trained diffusion models also work in purifying poisoning attacks."
> > - Yes, we are the first to apply EBMs and Diffusion models to purify train time poisons for classifier backdoors. In the paper, so as to not add extra complications (as we don't know how many of the diffusion models are trained on HuggingFace, if they use score matching etc) we trained our own simple DDPM diffusion model ourselves. In the original submitted paper we showed out of distribution(OOD) datasets work. Once you suggested to show numbers from per-trained HF models we simply ran the experiments you requested. But the results remained the same. OOD dataset trained diffusion/EMB models still work for purification. We are certainly the first to show this.
> >
> > "Considering the authors' claim that their methods provide SOTA defense, I have concerns about whether the advantages of specially designed techniques are still solid under more comprehensive comparisons, especially with more pre-trained diffusion models."
> > - We show a clear trade off between using in distribution and out of distribution data. We show that the user can choose to either train their own model or use a per-trained one. This gives the user freedom and confidence that even if they cannot muster up their own diffusion model, they should certainly purify their data with an EBM or diffusion model (in the schedule we suggest) to secure their dataset to SoTA levels.

---

### Official Review · Reviewer_gd22 · 2024-07-13

**Soundness:** 3
**Presentation:** 3
**Contribution:** 2
**Rating:** 6
**Confidence:** 4

**Summary:**

This work introduces PureGen, a method to purify a potentially poisoned dataset before the dataset is used to train a classifier. The work explores using Langevin MCMC with both EBMs and diffusion models to remove potential adversarial artifacts from data, with the justification that MCMC sampling should move a sample away from the adversarial artifacts and towards the data distribution. The number of Langevin steps is chosen to be enough to remove adversarial artifacts while preserving the original image. Experimental comparison shows strong performance of the proposed method for training robust classifier with a variety of data poisons compared to existing approaches.

**Strengths:**

* The presentation of the method is clear and straightforward, and the paper is easy to follow.
* The work proposes interesting adaptions of Langevin purification for white-box adversarial defense to the domain of data poisons.
* Empirical results show that the proposed defense can outperform existing defense. There is a thorough examination of different poisons and comparison with existing methods.

**Weaknesses:**

* The defense requires costly Langevin iterations and has a higher computational burden compared to existing methods. I appreciate that this is acknowledged clearly and discussed by the authors in the limitations section.
* My main concern about this work involves the situation where the data poisoner is aware of the defense method that will be applied to the dataset. Developing attacks that can adapt to different defense strategies is standard in the white-box attack literature, although I am not as familiar with such procedures in the poison literature. Will the defense remain robust if the attacker is aware of the defense strategy? See, for example, [a].

[a] https://arxiv.org/abs/1802.00420

**Questions:**

Could the authors discuss the situation where the attacker is aware of the defense? I am willing to raise my score if this concern can be addressed.

**Limitations:**

Limitations are adequately discussed.

---

> ### Author Rebuttal · Authors · 2024-08-07
>
> We thank the reviewer for their time and directly respond to the stated questions and weaknesses.
>
> ---
>
> ### Weaknesses
>
> 1. *The defense requires costly Langevin iterations and has a higher computational burden compared to existing methods.*
>
> The Langevin sampling is not as costly as it may seem. With modern GPUs, the cost is comparable to gradient descent. Our timings show that MCMC is actually much faster than the SoTA defense FRIENDS [3], as demonstrated in Table 5. This table also illustrates that **PureGen's cost becomes negligible when the purified dataset is reused.**
>
> 2. *Concern about the situation where the data poisoner is aware of the defense method that will be applied to the dataset and white-box scenarios.*
>
> In the context of train-time poisons (see general rebuttal **Train-Time vs. Inference-Time Attacks** for detailed differences), the poisoner must stealthily insert poisoned samples into the training dataset to create a backdoor in the NN. A central assumption is that the poisoner has no access to the training pipeline after impacting the train dataset, and the impact of those poisons remains undetected (minimal impact on standard train/test losses). Thus, in this context, **a white-box attack refers to the poisoner having the exact architecture, initialization, and weights (for transfer learning scenarios) of a pre-trained or from-scratch model, but they still lack access to the training pipeline and the model after training**. This scenario is addressed in Table 1, bottom right (Bullseye Polytope Linear Transfer White-Box Scenario), representing the strongest possible poison scenario.
>
> Although crafting poisons is outside the scope of defense papers, ***we did craft an EBM dynamics-aware Narcissus trigger and found that it was unable to poison against PureGen***. The stochastic nature of PureGen ensures that almost any effective poison perturbation will be high-energy and thus "purified" by PureGen dynamics, with the main concern being a tradeoff in natural accuracy.
>
> | Poison Craft Method                     |   Defense   |   Poison Success  |
> |:----------------------------------------|:-----------:|:-----------------:|
> | Narcissus Label 2 Baseline (In-Paper)   | No Defense  |      70.95 %      |
> | Narcissus Label 2 Baseline (In-Paper)   | PureGen-EBM |       2.70%       |
> | Narcissus Label 2 PureGen-EBM Aware     | No Defense  |       3.63%       |
> | Narcissus Label 2 PureGen-EBM Aware     | PureGen-EBM |       2.37 %      |
>
> Additional analysis of these results can be found in the general rebuttal **Defense-Aware Poison**. We encourage poison researchers to utilize our code and attempt to break the defense, but ***we already include results with the strongest scenarios available per the train-time poison literature along with this additional defense-aware experiment***. Note, crafting this single trigger took over 20+ hours of A100 GPU time. While we could not get more results for this rebuttal, we will include results and analysis in an appendix for the camera-ready version for all classes and for “PureGen-DDPM aware” Narcissus.
>
> ---
>
> ### Questions
>
> 1. *The reviewer asks if we could discuss defense-aware poisons.*
>
> See the Weakness 2 above and the **Defense-Aware Poison** section in the general rebuttal where we discuss this at length and include an additional experiment showing PureGen’s robustness to a custom defense-aware poison for PureGen-EBM.
>
> ---
>
> We hope these responses clarify any stated weaknesses and answer the reviewer’s questions. References are in general rebuttal. We look forward to any additional discussion.

---

> > ### Comment · Reviewer_gd22 · 2024-08-14
> > **Thanks for the response. I will raise my score.**
> >
> > I appreciate the efforts to investigate the defense-aware data poison, and it is reassuring to see the defense retains security. This addresses my main concern.
> >
> > While the EBM and diffusion purification methods fall within established practice, there appears to be a consensus among reviewers that this work does cover an area that has not yet been directly investigated in published work, namely purification-type defense applied for train-time data poisoning/purification. It does seem like there should be a work like this as a reference point for the community and the authors performed a wide array of experiments to investigate this scenario. I understand the concerns of other reviewers regarding novelty, but I nonetheless feel this work makes an expected but useful contribution. I will increase my score since my main concerns have been addressed.

---

### Official Review · Reviewer_16Y8 · 2024-07-19

**Soundness:** 3
**Presentation:** 3
**Contribution:** 1
**Rating:** 4
**Confidence:** 5

**Summary:**

This paper proposes a stochastic preprocessing defense technique, named PureGen, against train-time poisoning attacks, with EBM-Guided and Diffusion-Guided sampling processes. First, with EBM-based purification, called PureGen-EBM, the purifier first evaluates the (unnormalized) energy function of the images. Then, this considers the high-energy images as the 'posioned' images, and purifies with the stochastic preprocessor $\Psi_T$. On the other hand, with Diffusion-based purification, called PureGen-DDPM, the purifier first add Gaussian noises to the inputs and run the reverse diffusion process using DDPM.

**Strengths:**

* This paper demonstrates purification results with respect to diverse poisoning attacks and data availability attacks, which provides a new benchmarks on the adversarial learning communities.
* To the best of my knowledge, this is the first result to implement adversarial purification to poisoning attacks, achieving superior performances compared to other existing methods.

**Weaknesses:**

* The idea of using adversarial purification to adversarial attacks is already well-known, and this is an increment of the adversarial purification to other kinds of attacks.
* There are still scenarios that also breaks the adversarial robustness of the purification models such as BPDA+EOT [Athalye et al, 2018]. To mitigate this, additional consideration like fine-tuning on the adversarial perturbation models is required. [Lin et al., 2024]. Without this, the purified images will be easily poisoned with stronger poisoning attacks, which also involves with the purifier in the poisoning steps.
* Some additional validation on using the diffusion models on adversarial purification should be addressed: see __Questions__.
* In my opinion, with improved sampling methods using a higher-order solver of consistency-based distillation models, the purification can be drastically faster compared to both the proposed methods and the existing method. I presume that this can be easily improved: See __Questions__.

[Athalye et al, 2018] Obfuscated Gradients Give a False Sense of Security, ICML 2018 \
[Lin et al., 2024] Adversarial Training on Purification (AToP): Advancing Both Robustness and Generalization, ICLR 2024

**Questions:**

* In the PureGen-DDPM process, the dataset is trained with DDPM fewer steps, (250 steps rather than the conventional 1000 steps). Nevertheless, training DDPM with fewer discrete steps yields heavier posterior mismatch between the DDPM posterior covariance and the optimal covariance, according to [Bao et al, 2022]. I doubt that the purification gain from less timesteps from Figure 4 (bottom right) is just a side-effect of this posterior mismatch, which is not rigorously intended. I will be more convinced if some explanation on setting the fewer steps in PureGen-DDPM for training.
* Even though the time complexity of the proposed purification method is comparable to the existing methods, this can be easily improved with some higher-order solvers without any additional training. The paper will be much stronger if more results with some solvers are addressed.

[Bao et al, 2022] Analytic-DPM: an Analytic Estimate of the Optimal Reverse Variance in Diffusion Probabilistic Models, ICLR 2022

---

Minor typos
* (Appendix C) Intition $\to$ Intuition

---

> ### Author Rebuttal · Authors · 2024-08-07
>
> We thank the reviewer for their time and directly respond to the stated questions and weaknesses.
>
> ---
>
> ### Weaknesses
>
> 1. *Contributions are limited and adversarial purification is a known technique.*
>
> We respectfully disagree that our contribution is incremental and believe that conflating inference and train-time attacks contributes to this impression. In the general rebuttal (**Train-Time vs. Inference-Time Attacks**), we detail the differences between train and inference-time attacks. Train-time attacks require preventing the creation of "latent" backdoors during training, where defenders have no knowledge of poison presence in the data, and ***the attacker no longer has access to the training or inference pipeline after poisoning***. We demonstrate significant improvements across diverse train-time poisoning scenarios, establishing a new benchmark in this domain. Our extensive experiments provide practical applications beyond the norm in defense literature. We hope clarifying this distinction allows the reviewer to see the novelty of this work.
>
> 2. *There are still scenarios that break the adversarial robustness of the purification models such as BPDA+EOT [Athalye et al, 2018]. To mitigate this, additional consideration like fine-tuning on the adversarial perturbation models is required. [Lin et al., 2024].*
>
> **BPDA+EOT is an inference-time attack, which is out of scope for our paper.** PureGen is focused on train-time attacks, where attackers poison a dataset but lack access to the training pipeline afterwards. This central assumption in train-time attack and defense literature makes BPDA+EOT irrelevant here. As a side note, the SoTA defense against PGD+EOT(the full gradient version of BPDA+EOT) uses an EBM [1]. The published SoTAs for train-time poisons are EPIC [2] and FRIENDS [3], which do not defend against PGD/BPDA+EOT. DiffPure claims SoTA against BPDA+EOT but fails when true gradients, which is memory intensive and requires gradient checkpointing, are calculated i.e. PGD+EOT.
>
> **Fine-tuning in this way is not applicable in the train-time attack setting.** The train-time poisons in this paper are SoTA in strength, and our EBM and Diffusion models defend against them better than any other defense while maintaining natural accuracy. Even if future train-time poisons are stronger, the goals differ from PGD/BPDA+EOT and a comparison is not possible. For this rebuttal, ***we collected results crafting a defense-aware Narcissus patch (see **Defense-Aware Poison** in the general rebuttal), showing PureGen-EBM dynamics make crafting itself no longer possible in the standard budget*** (8/255) . Note, we are a defense paper, and not a poison paper, and coming up with a defense aware train time poison is not our burden, but we did so to answer these questions raised directly and indirectly by multiple reviewers.
>
> We provide concrete evidence that our PureGen method is SoTA under from-scratch, fine-tune, and linear transfer modes across various architectures, poison types, and attacker knowledge levels of gray and white-box scenarios covering all attacks relevant to train-time attack literature.
>
> ---
> ### Questions
>
> 1. *The reviewer asks about fewer DDPM steps in training and resulting heavier posterior mismatch*
>
> Our choice balances the fidelity of the original image with the need to effectively remove the poison perturbation and improve diffusion model training time. While a posterior mismatch is theoretically possible, **Figure 4 in Appendix B.3 shows empirical evidence that the truncated DDPM training schedule results in better poison defense and natural accuracy** for a given architecture and dataset. We will move this figure to the main paper due to its importance.
>
> We also include results with pre-trained Diffusion models (see **Usage of Pre-Trained Diffusion Models** in the general rebuttal), showing they can achieve reasonable defense success. However, for specific purification use-cases, the truncated PureGen-DDPM method offers improved performance and decreased training burden.
>
> 2. *The reviewer states even though the time complexity of the proposed purification method is comparable to the existing methods, this can be easily improved with some higher-order solvers without any additional training.*
>
> The time complexity of PureGen is significantly faster than SoTA defense FRIENDS as shown in Table 5, and is quickly marginalized over usage of the data as we only need to purify once. Using higher-order solvers can potentially give marginal speedups to diffusion, but as the reviewer mentioned, can be used on our diffusion models without re-training. **Since we only need to purify one time, the delta between standard DDMP and using a higher-order solver is negligible.** In the context of adversarial inference attacks where one must defend against every incoming image with gradients needing to be propagated through every step of diffusion, this may make a difference, but not in our train-time poison context.
>
> ---
>
> We hope these responses clarify any stated weaknesses and answer the reviewer’s questions. References are in general rebuttal. We look forward to any additional discussion.

---

> ### Comment · Reviewer_16Y8 · 2024-08-12
> **Response**
>
> Thank you for the detailed response and additional experiments. I agree with the following point, and would not reflect the following point for deducting the rating of the paper.
>
> __Purification on training-time attacks__
>
> I still consider that the novelty of this work is incremental, as directly applying diffusion/EBM (for adversarial purification) to mitigate poisoning attack. This would be the primary reason of my rating of this paper, unless having ample experiments.
>
> ---
>
> On the other hand, I address further questions below.
>
> __Defense-aware Poisoning__
>
> The authors mentioned that "_cannot hypothesize new poison techniques to bypass PureGen._" However, there is a simple and efficient poisoning method that can bypass the purification method.
> For example, we know from the literature that the ODE and SDE that runs through the diffusion model (and also the EBM) have the same density function, according to the _Fokker-Planck equation_.
>
> Then, for example, there can be a full pipeline that starts from the _purified image_ to the classifier:
> 1. begin with the noisy image, and run the purifier to obtain the clean(-ized) image.
> 2. Then, run the "poisoned" network for prediction.
>
> According to the authors' response, the authors presumed that the poisoning method for the purifier is beyond the scope. However, we can raise some issues:
>
>  - The ODE solver is effectively a series of deterministic forward neural network run. This means that, if we run 10 steps of ODE solver, the purifier-classifier pipeline can consist of __"10 steps of deterministic run"-"classifier"-"prediction"__, which is just a larger neural network. In this case, train-time poisoning is obviously available without complex methods bypassing the obfuscated gradients proposed in [Athalye et al., 2018] or so.
>
> Can the authors manage this issues that the attacker threatens not only the classifier but the generative models?
>
> ```
> [Athalye et al, 2018] Obfuscated Gradients Give a False Sense of Security, ICML 2018
> ```
>
> ---
>
> __Posterior Mismatch Issues__
>
> Firstly, we agree with the authors that this is not the primary issue of this paper, and __we do not deduct rating__ with respect to this following part. Nevertheless, there is still not enough relevance between [_Truncation of DDPM posterior_] and [_Better performance of train-time derense_] without any generic analysis. Hence, I do not agree with the authors to move this part into the main part of the paper.

---

> > ### Author Response · Authors · 2024-08-12
> >
> > We respectfully but firmly disagree with the reviewer's opinions.
> >
> > The reviewer states "This would be the primary reason of my rating of this paper, unless having ample experiments."
> >
> > How many experiments would be ample? We have trained over 3k classifiers throughout this paper showing that PureEBM and PureGen outperform all current defense methods which were published 2021/2022 at top conferences. In this we also trained 10s of 100s of EBMs and Diffusion models.
> >
> > The point of training poisons is that they are stealthy, as such the point of the reviewer that we could do SDEs to try to create a stronger attack is moot. To do this the attacker would need access to significantly more information than any of the poisoning papers are assuming.
> >
> > The reviewer also describes a scenario where one begins with a noisy image and then purifies and "Then, run the "poisoned" network for prediction." The reviewer is mistaken on the setup of poisons. This setup is for the adversarial attack and not poisons.
> >
> > The setup for from scratch poisons is this. No more no less.
> >
> > 1 . A poisoner adds small perturbations $\epsilon$ of their choosing to a small subset of images which will be used to train a classifier in step 2.
> >    - The poisoner can have access to at most the initialization of the classifier that will be trained in step 2.
> >
> > 2 . The train dataset that has been tampered with by the poisoner is used to train a classifier.
> >
> > 3 . If the poison is successful, the poisoner can pass a specific image of a cat through the classifier and have it be classified as a dog as they wanted.
> >
> > There is absolutely no other information given to either side. PGD attacks to augment step 3 are not allowed.
> >
> >
> > The reviewer states: "According to the authors' response, the authors presumed that the poisoning method for the purifier is beyond the scope."
> >
> > We feel this is a miss-characterization of what we have said. In fact trained the EBM/Diffusion models with 100% poisoned images. These poisons did not significantly affect our purification performance as seen in Table 3, where we considered fully
> > poisoned (all classes at once). The reviewer cites "Obfuscated Gradients Give a False Sense of Security, ICML 2018," again this paper is only on test time active attacks where the attacker has significantly more access to the classification problem, namely *fully trained model.* Again, this is a totally different problem and it not relevant to the setup of our paper. Still in the full PGD+EOT attack setup which is certainly outside of the cope of this paper, EBMs are still SOTA.

---

### Author Rebuttal · Authors · 2024-08-07

We thank the reviewers for their thoughtful feedback and the opportunity to address their concerns. Below, we provide a concise response to the main points raised by the reviewers and outline main revisions we will make to improve our paper.

---

### Responses to General Points

1. **Train-Time vs. Inference-Time Attacks**

***PureGen is the first method to use EBMs and DDPMs for defending against train-time data poisoning attacks.*** Our work focuses on train-time poisoning attacks, which aim to manipulate training data to cause misclassification during inference. This is distinct from inference-time attacks (PGD/BPDA+EOT), which aim to fool a trained model with perturbed test samples.

Clean-label train-time attackers manipulate a small portion of training data to introduce undetected backdoors, without further access to the model once training begins. In practice, attackers may disseminate poisoned images on public platforms like social media, hoping they are used for training. The goal is to avoid human detection, have images classified as intended in training, and not significantly alter training or validation metrics to avoid suspicion. We acknowledge works on generative models for inference-time attacks (e.g. Avatar [4], Stochastic Security [1]) but focus on the unique risk of train-time attacks, where an attacker creates a latent backdoor that can be exploited later without alerting the model trainer or deployer.

2. **List of Novel Contributions**:
- We are the first to bring generative model defense to train-time poisons and provide a comprehensive set of experiments showing SoTA performance over the strongest poisons and scenarios.
- We demonstrate that poisoned samples are separable as high-energy using EBMs, showing why generative model dynamics purify and lower the energy of such samples.
- We find generative models effective even when trained with distributionally shifted or poisoned data.
- We introduce a truncated DDPM training cycle to reduce computational costs and improve purification performance.
- We explore a combination of EBMs and DDPMs for better purification.

3. **Defense-Aware Poison**

To address concerns about “defense-aware” attackers, we conduct experiments using "defense-aware" Narcissus poisons crafted with knowledge of the EBM defense (using EBM dynamics in the crafting process directly). The results show PureGen's robustness, even against attacks designed with defense knowledge.

| Poison Craft Method                     |   Defense   |   Poison Success  |
|:----------------------------------------|:-----------:|:-----------------:|
| Narcissus Label 2 Baseline (In-Paper)   | No Defense  |      70.95 %      |
| Narcissus Label 2 Baseline (In-Paper)   | PureGen-EBM |       2.70%       |
| Narcissus Label 2 PureGen-EBM Aware     | No Defense  |       3.63%       |
| Narcissus Label 2 PureGen-EBM Aware     | PureGen-EBM |       2.37 %      |

Our “defense-aware” Narcissus experiments suggest two possible results: finding a hole in the EBM defense or failing to craft effective poisons due to stochastic purification. Our success against various poisons makes the first scenario unlikely. While not theoretically guaranteed, the new results further support that finding an effective poison with EBM knowledge is improbable. We can expand on this table for the camera-ready version but emphasize that we are not a poison paper and cannot hypothesize new poison techniques to bypass PureGen. We could not get more results for this rebuttal (crafting this single trigger took over 20+ hours of A100 GPU time), *we will include results and analysis for the camera-ready version for all classes*.


4. **Usage of Pre-Trained Diffusion Models**

We include an experiment using two pre-trained diffusion models from HuggingFace, showing similar results to our POOD DDPM results on Narcissus From-Scratch attack. PureGen-DDPM training is beneficial but not required for good defense performance; pre-trained models are often adequate when available.

|    | Model                  | Poison Success (%) |  Nat Acc (%) | Max Poison (%) |
|---:|:-----------------------|:------------------:|:------------:|:--------------:|
|  0 |PureGen-EBM CINIC-10_IN |    1.39 ± 0.80     | 92.92 ± 0.20 |      2.50     |
|  1 |PureGen-DDPM CINIC-10_IN|    1.64 ± 0.82     | 90.99 ± 0.22 |      3.83      |
|  2 |PureGen-DDPM Food-101   |    1.71 ± 0.74     | 88.35 ± 0.21 |      2.72      |
|  3 |PureGen-DDPM Office-Home|    1.80 ± 0.83     | 87.32 ± 0.22 |      3.16      |
|  4 |HuggingFace Butterflies [5] |    1.65 ± 0.83     | 87.79 ± 0.18 |      3.01      |
|  5 |HuggingFace Anime [6]    |    1.47 ± 0.75     | 90.91 ± 0.13 |      2.95      |

---

### Revision and Improvements

- We will include the results from both our additional experiments:
“Defense-aware” Narcissus showing PureGen is still effective when attackers can utilize the PureGen-EBM dynamics
Use of pre-trained Diffusion models that can obtain comparable defense performance to PureGen-DDPM
- We will clarify that the PureGen-DDPM training process shows gains empirically over standard diffusion but it not necessary to obtain reasonable defense performance
- We will add more detail to Section 1 to clearly enumerate our novel contributions.
- We will move Figure 4 into main paper

---

### References
[1] Hill M, Mitchell J, Zhu S. Stochastic Security: Adversarial Defense Using Long-Run Dynamics of Energy-based Models.
[2] Yang Y, Liu T Y, Mirzasoleiman B. Not All Poisons are Created Equal:  Robust Training against Data Poisoning. (EPIc).
[3] Liu T Y, Yang Y, Mirzasoleiman B. Friendly Noise against Adversarial Noise: A Powerful Defense against Data Poisoning Attacks.
[4] Dolatabadi H M, Erfani S, Leckie C. The devil’s advocate: Shattering the illusion of unexploitable data using diffusion models.
[5] https://huggingface.co/johnowhitaker/ddpm-butterflies-32px
[6] https://huggingface.co/onragi/anime-ddpm-32-res2-v3

---

### Decision · Program_Chairs · 2024-09-25

**Decision:**

Accept (poster)

**Comment:**

The reviews were split: On one hand, all reviewers acknowledged the novelty of this work as being the first to apply generative model based purification to training time poisoning attacks. On the other hand, Reviewers 16Y8 and n59K expressed concerns on the incremental nature of contributions, compared to existing purification work on inference attacks and availability attacks. Most other questions from the reviewers seem to have been largely addressed during the rebuttal.

While I agree with the reviewers that the idea of applying purification to mitigate training time poisoning attacks may sound natural and incremental, the authors have done commendable work to thoroughly verify and present the effectiveness of this idea. It is likely that many people in the field might benefit from learning these experimental results. And another tool to deal with data poisoning is certainly a welcome addition to the practitioners who have to deal with data in the wild and bear significant consequences had anything go awry. Thus, in the end I believe the merits and potential impact outweigh the concerns, and getting this work published sooner rather than later may encourage further, more significant efforts from the entire community.

Some suggestions for the next revision:

- Avoid using sentences like "people should be purifying all image data with whatever EBM/diffusion model they have access to," considering that there is certainly downsides and possible unknown side effects of purification.

- Make clear the poisoning attacks studied in this work are mostly targeted attacks, in contrast to indiscriminate attacks. Some of these were described in the appendix and it is important to clearly phrase the experimental setup as early as possible.

- Please incorporate the additional discussions and experiments during rebuttal.